# ESCAPING SADDLE POINT EFFICIENTLY IN MINIMAX AND BILEVEL OPTIMIZATIONS

## ABSTRACT

Hierarchical optimization is attracting significant attentions as it can be applied to a broad range of machine learning tasks. Recently, many algorithms are proposed to improve the theoretical results of minimax and bilevel optimizations. Among these works, a core issue that has not been well studies is to escape saddle point and find local minimum. In this paper, thus, we investigate the methods to achieve second-order optimality for nonconvex minimax and bilevel optimization. Specifically, we propose a new algorithm named PRGDA without the computation of second order derivative of the primal function. In nonconvex-strongly-concave minimax optimization, we prove that our algorithm can find a second-order stationary point with the gradient complexity that matches state-of-the-art result to find first-order stationary point. To our best knowledge, PRGDA is the first stochastic algorithm that is guaranteed to obtain the second-order stationary point for nonconvex minimax problems. In nonconvex-strongly-convex bilevel optimization, our method also achieves better gradient complexity to find local minimum. Finally, we conduct two numerical experiments to validate the performance of our new method.

## 1 INTRODUCTION

Hierarchical optimization (including minimax and bilevel optimization ) is a popular and important optimization framework which has been applied to a wide range of machine learning problems, such as Generative Adversarial Net (Goodfellow et al. (2014)), adversarial training (Madry et al. (2018)), multi-agent reinforcement learning (Wai et al. (2018)), meta-learning (Franceschi et al. (2018); Bertinetto et al. (2018)) and hyperparameter optimization (Shaban et al. (2019); Feurer & Hutter (2019)). In this paper, we study the following stochastic hierarchical optimization problem

$$\min_{x \in \mathbb{R}^{d_1}} \Phi(x) := f(x, y^*(x)) = \mathbb{E}_{\xi \in \mathcal{D}}[F(x, y^*(x); \xi)] \tag{1}$$

$$\text{s.t. } y^*(x) = \arg\min_{y \in \mathbb{R}^{d_2}} g(x, y) = \mathbb{E}_{\zeta \in \mathcal{D}'}[G(x, y; \zeta)],$$

where the upper-level function $f(x, y^*(x)) = \mathbb{E}_{\xi \in \mathcal{D}}[F(x, y^*(x); \xi)]$ is smooth and possibly nonconvex, and the lower-level function $g(x, y) = \mathbb{E}_{\zeta \in \mathcal{D}'}[G(x, y; \zeta)]$ is smooth and strongly-convex in variable $y$ so that $y^*(x)$ and $\Phi(x)$ can be well defined. $\xi$ and $\zeta$ are samples drawn from data distribution $\mathcal{D}$ and $\mathcal{D}'$. Stochastic problem is a general form that covers a couple of optimization tasks, including online optimization and finite-sum optimization. When $g(x, y) = -f(x, y)$, $\xi = \zeta$ and $\mathcal{D} = \mathcal{D}'$, the above hierarchical optimization (i.e., bilevel optimization) is reduced to a standard minimax optimization which can be rewritten as Eq. (2)

$$\min_{x \in \mathbb{R}^{d_1}} \max_{y \in \mathcal{Y} \subseteq \mathbb{R}^{d_2}} f(x, y) = \mathbb{E}_{\xi \in \mathcal{D}}[F(x, y; \xi)] \tag{2}$$

where $\mathcal{Y}$ is a convex domain (not required to be compact). The loss function $f(x, y)$ is smooth and possibly nonconvex w.r.t. $x$, and is smooth and strongly-concave w.r.t. $y$.

### 1.1 MINIMAX OPTIMZATION

Recently, there are plenty of works studying minimax optimization problem in a variety of research fields in machine learning. Many deterministic and stochastic algorithms with asymptotic or non-asymptotic convergence analysis have been developed, such as Gradient Descent Ascent (GDA) (Du & Hu (2019); Nemirovski (2004)) and Stochastic Gradient Descent Ascent (SGDA) (Lin et al. (2020a)). Some algorithms adopt a single loop structure (Heusel et al. (2017); Lin et al. (2020a);

Table 1: Comparison of properties between related algorithms for minimax optimization.

| Name | Reference | Stochastic | Local Minimum | Pure First-Order |
|---|---|---|---|---|
| SGDA | (Lin et al. (2020a)) | $\checkmark$ | $\times$ | $\checkmark$ |
| Cubic-GDA | (Chen et al. (2021b)) | $\times$ | $\checkmark$ | $\times$ |
| MCN | (Luo & Chen (2021)) | $\times$ | $\checkmark$ | $\times$ |
| Perturbed GDmax | (Huang et al. (2022b)) | $\times$ | $\checkmark$ | $\checkmark$ |
| PRGDA | (ours) | $\checkmark$ | $\checkmark$ | $\checkmark$ |

Xu et al. (2020)) while the others use a nested loop to update $y$ more frequently so that they can obtain a better estimation of the maximum $y^*(x)$ (Jin et al. (2019); Nouiehed et al. (2019)). Besides, some algorithms have been proposed to improve the theoretical results of minimax optimization, such as SREDA (Luo et al. (2020)) and Acc-MDA (Huang et al. (2022a)) which take advantage of variance reduction to accelerate the convergence rate and reduce the gradient complexity. Moreover, on deterministic setting some recently proposed algorithms (Lin et al. (2020b)) have already matched the optimal lower bound (Zhang et al. (2021)).

However, most of these works only consider the criterion of finding first-order stationary point. In nonconvex setting, convergence to first-order stationary point is not always satisfactory because a first-order stationary point could be a local minimum, saddle point or even local maximum. Therefore, second-order stationary point that reaches local minimum becomes a popular and important issue in nonconvex optimization. Since finding global minimum in nonconvex optimization is usually an NP-hard problem (Hillar & Lim (2013)), in some situations we attempt to find a local minimum instead. Moreover, in some machine learning tasks such as tensor decomposition (Ge et al. (2015)), matrix sensing (Bhojanapalli et al. (2016); Park et al. (2017)), and matrix completion (Ge et al. (2016)), finding local minimum is equivalent to finding global minimum, which makes second-order stationary point more crucial.

Therefore, we are motivated to study the method that obtains second-order stationary point for minimax (and bilevel) optimization which captures local minimum and escapes saddle point of $\Phi(x)$. In section 3 we can see that under certain conditions the objective function $\Phi(x)$ is twice differentiable and $\nabla^2\Phi(x)$ is Lipschitz continuous. An $O(\epsilon, \epsilon_H)$ second-order stationary point satisfies $\|\nabla\Phi(x)\| \leq O(\epsilon)$ and $\lambda_{min}(\nabla^2\Phi(x)) \geq -\epsilon_H$ where $\lambda_{min}(\cdot)$ means the smallest eigenvalue.

Although several recent works have been proposed to study the second-order stationary point for nonconvex-strongly-concave minimax optimization based on cubic-regularized gradient descent ascent (Chen et al. (2021b); Luo & Chen (2021)) or perturbed gradient (Huang et al. (2022b)), they are only adaptive to deterministic gradient oracle and finite-sum problem. The study of the second-order stationary point for stochastic nonconvex minimax problem where the full gradient is not available is still limited. A comparison of properties between related works for minimax optimization is demonstrated in Table 1.

Thus, to fill this gap, we propose a new algorithm named Perturbed Recursive Gradient Descent Ascent (PRGDA) to search second-order stationary point for stochastic nonconvex problem (2). To our best knowledge, PRGDA is the first algorithm that is guaranteed to obtain second-order stationary point for stochastic nonconvex minimax optimization problems. Furthermore, our method is a pure first-order algorithm that only requires the computation of gradient oracle. Neither Hessian matrix nor Hessian vector product is required, which makes our method more efficient to implement. We will also provide the analysis results to show that the gradient complexity of our algorithm is $\tilde{O}(\kappa^3\epsilon^{-3})$ to achieve $O(\epsilon, \sqrt{\rho_\Phi\epsilon})$ second-order stationary point where $\kappa$ is the condition number and $\rho_\Phi$ is the Lipschitz constant of $\nabla^2\Phi(x,y)$ (defined in section 3), which matches the best result of finding first-order stationary point for the same minimax optimization problem.

## 1.2 BILEVEL OPTIMIZATION

Recently, many algorithms have been studied to solve bilevel optimization. Some optimization algorithms are deterministic such as AID-BiO and ITD-BiO (Ji et al. (2021)) while the others consider stochastic algorithms including BSA (Ghadimi & Wang (2018)), TTSA (Hong et al. (2020)) and StocBiO (Ji et al. (2021)). These methods are proposed to improve the convergence analysis of bilevel optimization since most earlier works (Domke (2012); Pedregosa (2016)) only provide the asymptotic convergence analysis without specific convergence rates.

Table 2: Comparison of complexity between related algorithms for bilevel optimization. We use $p(\kappa)$ for some algorithms that do not provide the explicit dependence on $\kappa$.

| Name | Reference | $Gc(f, \epsilon)$ | $Gc(g, \epsilon)$ | Local Minimum |
|------|-----------|-------------------|-------------------|---------------|
| StocBiO | (Ji et al. (2021)) | $O(\kappa^5 \epsilon^{-4})$ | $O(\kappa^9 \epsilon^{-4})$ | $\times$ |
| SUSTAIN | (Khanduri et al. (2021)) | $O(p(\kappa)\epsilon^{-3})$ | $O(p(\kappa)\epsilon^{-3})$ | $\times$ |
| MRBO/VRBO | (Yang et al. (2021)) | $O(p(\kappa)\epsilon^{-3})$ | $O(p(\kappa)\epsilon^{-3})$ | $\times$ |
| StocBiO + iNEON | (Huang et al. (2022b)) | $\tilde{O}(\kappa^5 \epsilon^{-4})$ | $\tilde{O}(\kappa^{10} \epsilon^{-4})$ | $\checkmark$ |
| PRGDA | (ours) | $\tilde{O}(\kappa^3 \epsilon^{-3})$ | $\tilde{O}(\kappa^7 \epsilon^{-3})$ | $\checkmark$ |

StocBiO algorithm (Ji et al. (2021)) is a recent work to solve stochastic nonconvex-strongly-convex bilevel optimization via AID. In this paper, we also study the convergence of our method under this condition where $\Phi(x)$ is stochastic and probably nonconvex. According to previous studies of bilevel optimization, when $f(x, y)$ and $g(x, y)$ are differentiable and $g(x, y)$ is strongly-convex with respect to $y$, $\Phi(x)$ is also differentiable and automatically $\|\nabla\Phi(x)\| \leq \epsilon$ is a criterion of first-order stationary point. Notice that in (Ji et al. (2021)) $\|\nabla\Phi(x)\|^2 \leq \epsilon$ is used as the criterion. In this paper, we will uniformly adopt $\|\nabla\Phi(x)\| \leq \epsilon$ as the convergence criterion. More recently, many stochastic algorithms with variance reduction are proposed, such as RSVRB (Guo & Yang (2021)), SUSTAIN (Khanduri et al. (2021)), MRBO and VRBO (Yang et al. (2021)). The gradient complexity of bilevel optimization is enhanced to $O(\epsilon^{-3})$, which is the best theoretical result as far as we know. StocBiO with iNEON (Huang et al. (2022b)) is another recent work that combines StocBiO algorithm with pure first-order method inexact negative curvature originated from noise (iNEON) to escape saddle point and find second-order stationary point for nonconvex-strongly-convex bilevel optimization.

Although these works are proposed to improve the performance of algorithms for bilevel optimization, the complexity of current methods that achieve second-order stationary point are still high. Actually, the complexity of StocBiO with iNEON is even higher than the standard StocBiO algorithm in order to find a local minimum with high probability. Thus, to fill these gap, we are motivated to propose an accelerated algorithm with variance reduction that requires lower complexity to find second-order stationary point for stochastic nonconvex-strongly-convex bilevel optimization.

The comparison of gradient complexity between our method and related works to find $O(\epsilon)$ first-order stationary point or $O(\epsilon, \sqrt{\rho_\Phi \epsilon})$ second-order stationary point is shown in Table 2. In Table 2, $Gc(f, \epsilon)$ and $Gc(g, \epsilon)$ are the numbers of gradient evaluations of function $f(x, y)$ and $g(x, y)$ respectively. The last column represents whether the algorithm is able to escape saddle point and find local minimum. Notation $\tilde{O}$ hides the logarithm term. StocBiO with iNEON and our PRGDA algorithm involve a logarithm term in the complexity because they converge to second-order stationary point with high probability, considering all randomness including the stochastic gradient while other algorithms only consider the expectation over stochastic gradients. From Table 2 we can see our PRGDA algorithm improves the gradient complexity $Gc(f, \epsilon)$ and $Gc(g, \epsilon)$ of StocBiO with iNEON algorithm significantly and matches state-of-the-art complexity $O(\epsilon^{-3})$, which is one of the most important contribution of this paper.

### 1.3 CONTRIBUTIONS

We summarize our main contributions as follows:

- We propose a new PRGDA algorithm which is the first algorithm to reach second-order stationary point for stochastic nonconvex minimax optimization problem. Our method is pure first-order and does not require any calculation of second-order derivatives. Our method does not involve nested loops either, which makes it more efficient to implement.

- We prove that the gradient complexity of our algorithm is $\tilde{O}(\kappa^3 \epsilon^{-3})$ to achieve $O(\epsilon, \sqrt{\epsilon})$ second-order stationary point in stochastic nonconvex minimax optimization, which matches the best result of finding first-order stationary point in the same problem.

- Our PRGDA algorithm can also be applied to nonconvex bilevel optimization and we can prove that the gradient complexity is $Gc(f, \epsilon) = \tilde{O}(\kappa^3 \epsilon^{-3})$ and $Gc(g, \epsilon) = \tilde{O}(\kappa^7 \epsilon^{-3})$ to find $O(\epsilon, \sqrt{\epsilon})$ second-order stationary point in stochastic nonconvex bilevel optimization, which outperforms the previous best theoretical results and matches state-of-the-art to find first-order stationary point.

## 2 RELATED WORK

In this section we will summarize the background of related works and some details of methods that are important to our work will be further discussed in the Appendix.

## 2.1 STOCHASTIC MINIMAX OPTIMIZATION

Many algorithms are proposed to solve stochastic nonconvex-strongly-concave minimax problem, including intuitive methods SGDmax (Jin et al. (2019)) and Stochastic Gradient Descent Ascent (SGDA) (Lin et al. (2020a)). More recently in (Yang et al. (2022)), a new method Stoc-Smoothed-AGDA is proposed to achieve better complexity with a weaker PL condition instead of strong concavity. Besides, some methods integrate variance reduction with minimax problem to accelerate the convergence, such as Stochastic Recursive gradiEnt Descent Ascent (SREDA) (Luo et al. (2020)), Hybrid Variance-Reduced SGD (Tran-Dinh et al. (2020)) and Acc-MDA (Huang et al. (2022a)). There are also some works that study the weakly-convex concave minimax optimization such as (Rafique et al. (2021)) and (Yan et al. (2020)). More related to this work, Cubic-Regularized Gradient Descent Ascent (Cubic-GDA) (Chen et al. (2021b)) and Minimax Cubic Newton (MCN) (Luo & Chen (2021)) are two recent algorithms that can reach the second-order stationary point in nonconvex-strongly-concave minimax optimization.

## 2.2 PERTURBED GRADIENT DESCENT

Perturbed Gradient Descent (PGD) (Jin et al. (2017)) was proposed to find second-order stationary point for nonconvex optimization which introduces a perturbation under specific condition. It is a deterministic gradient based algorithm and only involves first-order oracle. To extend Perturbed Gradient Descent to the stochastic setting and incorporate it with variance reduction, SSRGD Li (2019) was proposed to reach second-order stationary point with SFO of $O(\epsilon^{-3.5})$. After that Pullback algorithm (Chen et al. (2021a)) was proposed to improve the complexity to $O(\epsilon^{-3})$.

## 2.3 STOCBIO WITH INEON

In (Huang et al. (2022b)), algorithms for both minimax and bilevel optimization are proposed to find second-order stationary point. However, for minimax optimization only the deterministic problem is studied. In the proposed Perturbed GDmax algorithm, perturbed gradient descent is used to solve the issue in this case. As we have mentioned, perturbed gradient descent in deterministic and stochastic are totally different. Therefore, it is essential to investigate the stochastic minimax optimization algorithm that converge to second-order stationary point. For bilevel optimization, the stochastic problem is considered and the StocBiO with iNEON algorithm is proposed. The algorithm is inspired by NEON (Xu et al. (2018); Allen-Zhu & Li (2018)), which is a method to find local minimum merely based on first-order oracles. Inexact NEON is a variant of NEON since the exact gradient in bilevel optimization is unavailable. However, it requires an extra nested loop to solve a subproblem that extracts a negative curvature descent direction. Besides, the gradient complexity of StocBiO with iNEON is also higher than the vanilla StocBiO. Therefore, we are motivated to propose a more efficient bilevel optimization algorithm that converges to second-order stationary point.

## 3 PRELIMINARY

In this section we will present the notations used in this paper and introduce some basic assumptions to further illustrate the problem setting. We assume that upper-level function $f(x, y)$ is twice differentiable. Lower-level $g(x, y)$ is three times differentiable (only required in bilevel optimization). The partial derivative is denoted by $\nabla_x$ and $\nabla_y$, e.g., $\nabla f(x, y) = [\nabla_x f(x, y), \nabla_y f(x, y)]$. Similarly, $\nabla_x^2$ and $\nabla_y^2$ represent the Hessian. $\nabla_{xy}^2$ and $\nabla_{yx}^2$ represent the Jacobian. We use $\|\cdot\|_2$ and $\|\cdot\|_F$ to denote the spectral norm and Frobenius norm of matrix respectively. Notation $\tilde{O}(\cdot)$ means the complexity after hiding logarithm terms. First, we assume that lower-level function $g(x, y)$ is strongly-convex with respect to $y$ so that $y^*(x)$ and $\Phi(x)$ can be well defined.

**Assumption 1.** *The lower-level function $g(x, y)$ is $\mu$-strongly-convex with respect to $y$, i.e., there exists a constant $\mu$ such that*

$$g(x, y) + \langle \nabla_y g(x, y), y' - y \rangle + \frac{\mu}{2} \|y' - y\|^2 \leq g(x, y') \tag{3}$$

*for any $x$, $y$ and $y'$.*

Notice that in minimax optimization $g(x, y)$ is the same as $-f(x, y)$ so we merge these two cases into one statement. With Assumption 1, objective function $\Phi(x)$ is also differentiable and the gradient is formulated as follows (Ji et al. (2021))

$$\nabla \Phi(x) = \nabla_x f(x, y^*(x)) - \nabla_{xy}^2 g(x, y^*(x))[\nabla_y^2 g(x, y^*(x))]^{-1} \nabla_y f(x, y^*(x)) \tag{4}$$

We can see the Hessian of $g$ is automatically involved in the gradient of $\Phi$. **Notice** that in this paper first-order method means only using the first-order information of $\Phi$. In minimax optimization, since we always have $\nabla_y f(x, y^*(x)) = 0$, the expression of $\nabla \Phi(x)$ is simplified by

$$\nabla \Phi(x) = \nabla_x f(x, y^*(x)) \tag{5}$$

Next, we introduce the following assumptions about Lipschitz continuity of first and second order derivatives. These assumptions are commonly used in the convergence analysis of minimax and bilevel optimization (Luo et al. (2020); Luo & Chen (2021); Ji et al. (2021); Huang et al. (2022b)).

**Assumption 2.** *The gradients of component functions $F(x, y; \xi)$ and $G(x, y; \zeta)$ are $L$-Lipschitz continuous, i.e., there exists a constant $L$ such that*

$$\|\nabla F(z; \xi) - \nabla F(z'; \xi)\| \leq L\|z - z'\|, \quad \|\nabla G(z; \zeta) - \nabla G(z'; \zeta)\| \leq L\|z - z'\| \tag{6}$$

*for any $z = (x, y)$ and $z' = (x', y')$.*

**Assumption 3.** *The second order derivatives $\nabla_x^2 f(x, y)$, $\nabla_{xy}^2 f(x, y)$, $\nabla_y^2 f(x, y)$, $\nabla_{xy}^2 g(x, y)$ and $\nabla_y^2 g(x, y)$ are $\rho$-Lipschitz continuous.*

The condition number $\kappa$ of the hierarchical optimization problem is defined by $\kappa = L/\mu$. According to previous works, in minimax optimization under Assumptions 1, 2 and 3, $\Phi(x)$ is twice differentiable. $y^*(x)$ is $\kappa$-Lipschitz continuous, $\nabla \Phi(x)$ is $L_\Phi$-Lipschitz continuous and $\nabla^2 \Phi(x)$ is $\rho_\Phi$-Lipschitz continuous. According to (Ghadimi & Wang (2018); Ji et al. (2021)), we know in bilevel optimization function $y^*(x)$ is also $\kappa$-Lipschitz continuous, but we need an additional Assumptions 4 to guarantee $\Phi(x)$ has $L_\Phi$-Lipschitz gradient.

**Assumption 4.** *The upper-level function $f(x, y)$ is $M$-Lipschitz continuous, i.e., there exists a constant $M$ such that*

$$\|f(z) - f(z')\| \leq M\|z - z'\| \tag{7}$$

*for any $z = (x, y)$ and $z' = (x', y')$.*

Since in this paper we study the convergence to second-order stationary point, we also need the following Assumption 5 which is also assumed in (Huang et al. (2022b)) that makes function $\Phi(x)$ twice differentiable and have $\rho_\Phi$-Lipschitz Hessian. We should notice that Assumption 4 and 5 are **only** used for bilevel optimization.

**Assumption 5.** *The third order derivatives $\nabla_{xyx}^3 g$, $\nabla_{yxy}^3 g$ and $\nabla_y^3 g$ are $\nu$-Lipschitz continuous.*

## 4 PROPOSED ALGORITHM FOR MINIMAX OPTIMIZATION

In this section, we will propose our PRGDA algorithm for the special case of minimax optimization. The description of our PRGDA algorithm is demonstrated in Algorithm 1. Similar to SREDA, the initial value $y_0$ is also yield by PiSARAH algorithm to make it close to $y^*(x_0)$, which is a conventional strongly-convex optimization subproblem. In our convergence analysis this step costs the gradient complexity of $\tilde{O}(\kappa^2\epsilon^{-2})$. We use $v_t$ and $u_t$ to represent the gradient estimator of $\nabla_x f(x_t, y_t)$ and $\nabla_y f(x_t, y_t)$ respectively. In each iteration, $y_{t+1}$, $v_t$ and $u_t$ are computed by an inner loop updater with $K$ iterations, which is shown in Algorithm 2. In Algorithm 2, we use the SPIDER gradient estimator to update $y_{t,k}$, $v_{t,k}$ and $u_{t,k}$. $S_1$ is the large batchsize that is loaded every $q$ iterations of $t$. $S_2$ is the small batchsize. $\lambda$ is the stepsize to update variable $y$. The output of the inner loop updater depends on the minimum value of the norm of $\tilde{\mathcal{G}}_\lambda(y_{t,k})$ and its corresponding index, which is defined by $\tilde{\mathcal{G}}_\lambda(y_{t,k}) = (y_{t,k} - \Pi_{\mathcal{Y}}(y_{t,k} + \lambda u_{t,k}))/\lambda$. We will show that gradient estimator $v_t$ satisfies $\|v_t - \nabla \Phi(x_t)\| \leq O(\epsilon)$ based on this inner loop updater.

Inspired by perturbed gradient descent, our PRGDA is also composed of a descent phase and an escaping phase. In the descent phase our PRGDA algorithm follows the iterative update rule of SPIDER that $x_{t+1} = x_t - (\eta/\|v_t\|)v_t$ until the norm of $v_t$ satisfies $\|v_t\| \leq O(\epsilon)$. After the descent phase is terminated, we use $m_s$ to denote the current counter $t$ and uniformly draw a perturbation $\xi$ from ball $B_0(r)$ where parameter $r$ is the perturbation radius. We add the perturbation to the current status $x_t$ and start the escaping phase. In the escaping phase, parameter $t_{thres}$ is maximum number of iterations of the phase and $\bar{D}$ is the average moving distance which is used to determine if the escaping phase should be stopped. The stepsize of $x$ in this phase is denoted by $\eta_H$ which is typically larger than $\eta$ in the descent phase. We use $D$ to denote the accumulated squared moving distance. If the averaged squared moving distance is larger than $\bar{D}$ then we pull it back (line 17 in Algorithm 1) and break the escaping phase. In this case we consider $x_{m_s}$ as a saddle point and continue to run next descent phase. Otherwise, if the escaping phase is not broken after $t_{thres}$ iterations, we claim that $x_{m_s}$ is a second-order stationary point with high probability. This is because when $\lambda_{min}(\nabla^2 \Phi(x_{m_s})) < -\epsilon_H$, the stuck region $\mathcal{S}$ defined by the area

---

**Algorithm 1** Perturbed Recursive Gradient Descent Ascent

---

**Input**: initial value $x_0, y_0$
**Parameter**: stepsize $\eta$ and $\eta_H$, perturbation radius $r$, escaping phase threshold $t_{thres}$, average movement $\bar{D}$, tolerance $\epsilon$, maximum iteration $T$.

1: Set $escape = false$, $s = 0$, $esc = 0$.
2: **for** $t = 0, 1, \ldots, T-1$ **do**
3:     Update $y_{t+1}$, $v_t$, $u_t$ from Algorithm 2 (**Minimax**) or Algorithm 3 (**Bilevel**).
4:     **if** $escape = false$ **then**
5:       **if** $\|v_t\| \geq \epsilon$ **then**
6:         Update $x_{t+1} = x_t - (\eta/\|v_t\|)v_t$.
7:       **else**
8:         Let $m_s = t$, $s = s+1$, $escape = true$, $esc = 0$.
9:         Draw perturbation $\xi \sim B_0(r)$ and update $x_{t+1} = x_t + \xi$.
10:       **end if**
11:     **else**
12:       Compute $D = \sum_{j=m_s+1}^{t} \eta_H^2 \|v_j\|^2$.
13:       **if** $D > (t - m_s)\bar{D}$ **then**
14:         Set $\eta_t$ s.t. $\sum_{j=m_s+1}^{t} \eta_j^2 \|v_j\|^2 = (t - m_s)\bar{D}$.
15:         Update $x_{t+1} = x_t - \eta_t v_t$. Set $escape = false$.
16:       **else**
17:         Set $\eta_t = \eta_H$. Update $x_{t+1} = x_t - \eta_t v_t$, $esc = esc + 1$.
18:         **Return** $x_{m_s}$ **if** $esc = t_{thres}$.
19:       **end if**
20:     **end if**
21: **end for**
**Output**: $x_{m_s}$

---

$\{\xi \in B_0(r)|$ the sequence started from $x_{m_s+1} = x_{m_s} + \xi$ does not break the escaping phase$\}$ has a small volume. Specifically, similar to Lemma 6 in (Li (2019)) and Lemma D.3 in (Chen et al. (2021a)), we can prove if we suppose after the perturbation there are two coupled sequences started from two points $x_{m_s+1}$ and $x'_{m_s+1}$ respectively within a small distance $\|x_{m_s+1} - x'_{m_s+1}\| = r_0$ in the smallest eigenvector direction of Hessian matrix $\nabla^2 \Phi(x_{m_s})$, then there must be at least one sequence $\{x_{m_s+1}\}$ or $\{x'_{m_s+1}\}$ that breaks the escaping phase. Informally, this means the stuck region $\mathcal{S}$ must be contained in a "narrow band" or "thin disk" in a high dimensional space which cannot have a large measure. Since the perturbation $\xi$ is uniformly generated from ball $B_0(r)$, the probability that $\xi$ belongs to the stuck region is low.

## 5 PROPOSED ALGORITHM FOR BILEVEL OPTIMIZATION

In this section we propose our PRGDA algorithm to solve the more general bilevel optimization. Actually, we only need to switch the inner loop updater in Algorithm 2 to the bilevel mode, which is demonstrated in Algorithm 3 in Appendix. Similar to the case of minimax optimization, here we also need a initialization algorithm to initialize $y_0$ with the cost of $Gc(g, \epsilon) = \tilde{O}(\kappa^6 \epsilon^{-2})$ in the convergence analysis. Next we will elaborate the inner loop updater for bilevel optimization. We also use the update rule of SPIDER to compute $v_{t,k}^{(1)}$, $v_{t,k}^{(2)}$ and $u_{t,k}$, which represent the estimator of $\nabla_x f(x,y)$, $\nabla_y f(x,y)$ and $\nabla_y g(x,y)$ respectively. We should notice that the large and small batchsize of computing $u_{t,k}$ are different from that of $v_{t,k}^{(1)}$ or $v_{t,k}^{(2)}$. After the inner loop to compute $y_{t+1}$, we calculate the Jacobian $J_t$ with a batch of size $S_5$. Then we compute $v_t$, the estimator of $\nabla\Phi(x)$ via AID. Here we follow the method used in StocBiO, which is

$$z_t^Q = \alpha \sum_{q=-1}^{Q-1} \prod_{j=Q-q}^{Q} (I - \alpha\nabla_y^2 G(x_t, y_{t+1}; \mathcal{B}_j))v_t^{(2)}, \quad v_t = v_t^{(1)} - J_t z_t^Q \tag{8}$$

where $\mathcal{B}_j$ is the set of samples to calculate the stochastic estimator of Hessian $\nabla_y^2 g(x_t, y_{t+1})$.

## 6 CONVERGENCE ANALYSIS

In this section we will illustrate the main theorem and provide the convergence analysis of our algorithm. First, we need to assume that $\Phi(x)$ is lower bounded by $\Phi^*$. Then we will present the

---

**Algorithm 2** Updater of Inner Loop (Minimax)

---

**Input**: status $x_t$, $x_{t-1}$, $y_t$, $v_{t-1}$, $u_{t-1}$ and $t$
**Parameter**: stepsize $\lambda$, inner loop size $K$, batchsize $S_1$ and $S_2$, period $q$.

1: Set $x_{t,-1} = x_{t-1}$, $x_{t,k} = x_t$ when $k \geq 0$, $y_{t,-1} = y_{t,0} = y_t$.
2: **if** $mod(t,q) = 0$ **then**
3:     Draw $S_1$ samples $\{\xi_1, \ldots, \xi_{S_1}\}$
4:     Compute $v_{t,-1} = \frac{1}{S_1}\sum_{i=1}^{S_1}\nabla_x F(x_t, y_t; \xi_i)$, $u_{t,-1} = \frac{1}{S_1}\sum_{i=1}^{S_1}\nabla_y F(x_t, y_t; \xi_i)$.
5: **else**
6:     Let $v_{t,-1} = v_{t-1}$, $u_{t,-1} = u_{t-1}$.
7: **end if**
8: **for** $k = 0$ to $K-1$ **do**
9:     Draw $S_2$ samples $\{\xi_1, \ldots, \xi_{S_2}\}$
10:     Compute $v_{t,k} = v_{t,k-1} + \frac{1}{S_2}\sum_{i=1}^{S_2}(\nabla_x F(x_{t,k}, y_{t,k}; \xi_i) - \nabla_x F(x_{t,k-1}, y_{t,k-1}; \xi_i))$
11:     Compute $u_{t,k} = u_{t,k-1} + \frac{1}{S_2}\sum_{i=1}^{S_2}(\nabla_y F(x_{t,k}, y_{t,k}; \xi_i) - \nabla_y F(x_{t,k-1}, y_{t,k-1}; \xi_i))$
12:     $y_{t,k+1} = \prod_{\mathcal{Y}}(y_{t,k} + \lambda u_{t,k})$.
13: **end for**
14: Select $s_t = \arg\min_k \|\tilde{\mathcal{G}}_\lambda(y_{t,k})\|$. Let $y_{t+1} = y_{t,s_t}$, $v_t = v_{t,s_t}$, $u_t = u_{t,s_t}$.
**Output**: $y_{t+1}$, $v_t$, $u_t$.

---

main theorems of our PRGDA algorithm. In this paper, we set $\epsilon_H = \sqrt{\rho_\Phi \epsilon}$ as the tolerance of the second-order stationary point. We leave the proof of Theorem 1 and 2 to the Appendix.

### 6.1 MAIN THEOREM FOR MINIMAX OPTIMIZATION

**Theorem 1.** *Under Assumption 1, 2 and 3, we set stepsize $\eta = \tilde{O}(\frac{\epsilon}{\kappa L})$, $\eta_H = \tilde{O}(\frac{1}{\kappa L})$ and $\lambda = O(\frac{1}{L})$, batchsize $S_1 = \tilde{O}(\kappa^2\epsilon^{-2})$ and $S_2 = \tilde{O}(\kappa\epsilon^{-1})$, period $q = O(\epsilon^{-1})$, inner loop $K = O(\kappa)$, perturbation radius $r = \min\{\tilde{O}(\sqrt{\frac{\epsilon}{\kappa^3\rho}}), \tilde{O}(\frac{\epsilon}{\kappa L})\}$, threshold $t_{thres} = \tilde{O}(\frac{L}{\sqrt{\kappa\rho\epsilon}})$ and average movement $\bar{D} = \tilde{O}(\frac{\epsilon^2}{\kappa^2 L^2})$. Then our PRGDA algorithm requires $\tilde{O}(\kappa^3\epsilon^{-3})$ SFO complexity to achieve $O(\epsilon, \sqrt{\rho_\Phi\epsilon})$ second-order stationary point with high probability.*

### 6.2 MAIN THEOREM FOR BILEVEL OPTIMIZATION

**Theorem 2.** *Under Assumption 1, 2, 3, 4 and 5, we set stepsize $\eta = \tilde{O}(\frac{\epsilon}{\kappa^3 L})$, $\eta_H = \tilde{O}(\frac{1}{\kappa^3 L})$, $\lambda = O(\frac{1}{L})$ and $\alpha = O(\frac{1}{L})$, batchsize $S_1 = \tilde{O}(\kappa^2\epsilon^{-2})$, $S_2 = \tilde{O}(\kappa^{-1}\epsilon^{-1})$, $S_3 = \tilde{O}(\kappa^6\epsilon^{-2})$, $S_4 = \tilde{O}(\kappa^3\epsilon^{-1})$, $S_5 = \tilde{O}(\kappa^2\epsilon^{-2})$ and $B = \tilde{O}(\kappa^2\epsilon^{-2})$, period $q = O(\kappa^2\epsilon^{-1})$, inner loop $K = O(\kappa)$ and $Q = \tilde{O}(\kappa)$, perturbation radius $r = \min\{\tilde{O}(\sqrt{\frac{\epsilon}{\rho_\Phi}}), \tilde{O}(\frac{\epsilon}{\kappa^3 L})\}$, threshold $t_{thres} = \tilde{O}(\frac{\kappa^3 L}{\sqrt{\rho_\Phi\epsilon}})$ and average movement $\bar{D} = \tilde{O}(\frac{\epsilon^2}{\kappa^6 L^2})$. Then our PRGDA algorithm requires complexity of $Gc(f, \epsilon) = \tilde{O}(\kappa^3\epsilon^{-3})$, $Gc(g, \epsilon) = \tilde{O}(\kappa^7\epsilon^{-3})$, $JV(g, \epsilon) = \tilde{O}(\kappa^5\epsilon^{-4})$ and $HV(g, \epsilon) = \tilde{O}(\kappa^6\epsilon^{-4})$ to achieve $O(\epsilon, \sqrt{\rho_\Phi\epsilon})$ second-order stationary point with high probability.*

## 7 EXPERIMENTS

In this section we conduct the matrix sensing (Bhojanapalli et al. (2016); Park et al. (2017)) experiment to validate the performance of out PRGDA algorithm for solving both minimax and bilevel problem. As a result of existing study on matrix sensing problem (Ge et al. (2017)), there is no spurious local minimum in this circumstance, *i.e.*, every local minimum is a global minimum. Therefore, the capability of escaping saddle points of our algorithm can be verified by this experiment. We follow the experiment setup of (Chen et al. (2021a)) to recover a low-rank symmetric matrix $M^* = U^*(U^*)^T$ where $U^* \in \mathbb{R}^{d \times r}$. Suppose we have $n$ sensing matrices $\{A_i\}_{i=1}^n$ with $n$ observations $b_i = \langle A_i, M^* \rangle$. Here the inner product of two matrices is defined by the trace $\langle X, Y \rangle = tr(X^T Y)$. Then the optimization problem can be defined by

$$\min_{U \in \mathbb{R}^{d \times r}} \frac{1}{2}\sum_{i=1}^n L_i(U), \ L_i(U) = (\langle A_i, UU^T \rangle - b_i)^2 \tag{9}$$

The code of our algorithms is uploaded in the Supplementary Material.

### 7.1 ROBUST OPTIMIZATION

Similar to the problem setting of (Yan et al. (2019)), we also introduce another variable $y$ and add a robust term to make the model robust. Therefore, the optimization problem can be formulated by

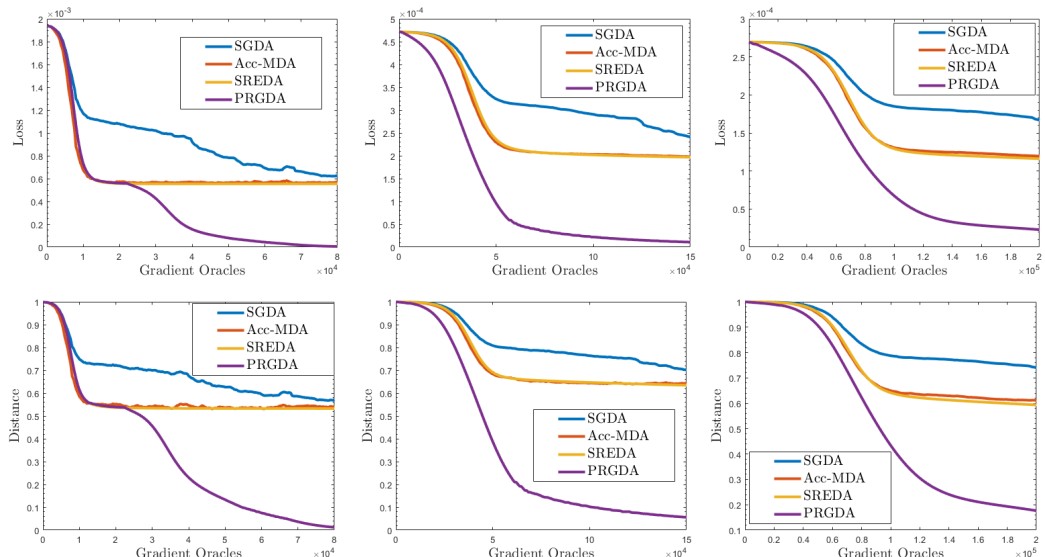

Figure 1: Experimental results of our robust low-rank matrix sensing task. Figure (a) to (c) show the loss function value of $\Phi(U)$ against the number of gradient oracles with $d = 50$, $d = 75$, and $d = 100$ respectively. Figure (d) to (f) show the ratio of distance $\|UU^T - M^*\|_F^2 / \|M^*\|_F^2$ against the number of gradient oracles with $d = 50$, $d = 75$, and $d = 100$ respectively.

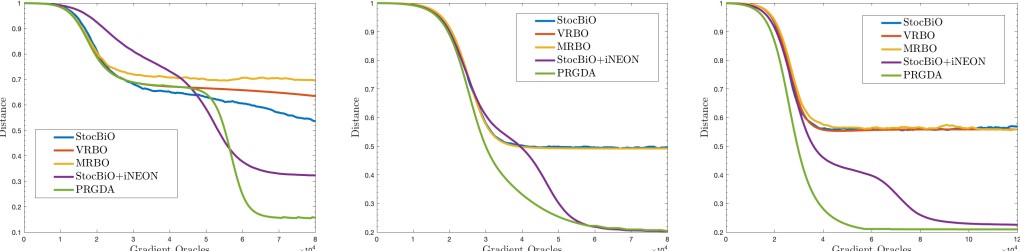

Figure 2: Experimental results of our hyper-representation learning of low-rank matrix sensing task. The ratio of distance $\|UU^T - M^*\|_F^2 / \|M^*\|_F^2$ is shown against the number of gradient oracles with $d = 50$, $d = 75$, and $d = 100$ respectively.

$$\min_{U \in \mathbb{R}^{d \times r}} \max_{y \in \Delta_n} f(U, y) = \frac{1}{2} \sum_{i=1}^{n} y_i L_i(U) - (y_i - \frac{1}{n})^2 \tag{10}$$

where $\Delta_n = \{y \in \mathbb{R}^n | 0 \le y_i \le 1, \sum_{i=1}^{n} y_i = 1\}$ is the simplex in $\mathbb{R}^n$ and $L_i(U)$ is defined in Eq. (9). The number of rows of matrix $U$ is set to $d = 50$, $d = 75$ and $d = 100$ respectively and the number of columns is fixed as $r = 3$ in the main manuscript. The results of different ranks will be shown in the Appendix. The ground truth low-rank matrix $M^*$ is generated by $M^* = U^*(U^*)^T$ where each entry of $U^*$ is drawn from Gaussian distribution $\mathcal{N}(0, 1/d)$ independently. We randomly generate $n = 20d$ samples of sensing matrices $\{A_i\}_{i=1}^{n}$, $A_i \in \mathbb{R}^{d \times d}$ from standard Gaussian distribution and calculate the corresponding labels $b_i = \langle A_i, M^* \rangle$ hence there is no noise in the synthetic data. The global minimum of loss function value $\Phi(U)$ should be 0 which can be achieved at point $U = U^*$ and $y = \mathbf{1}/n$.

Following the setup in (Chen et al. (2021a)), we randomly generalize a vector $u_0$ from Gaussian distribution and multiply it by a scalar such that it satisfies the condition $\|u_0\| \le \lambda_{max}(M^*)$ where we denote $\lambda_{max}(\cdot)$ as the maximum eigenvalue. The initial value is set to $U = [u_0, \mathbf{0}, \mathbf{0}]$. Each optimization algorithm shares the same initialization. Apart from our PRGDA algorithm, we run three baseline algorithms, SGDA, Acc-MDA and SREDA. The code is implemented on matlab. We choose $\eta = 0.001$, $\eta_H = 0.1$, $\lambda = 0.01$, $\bar{D} = r = 0.01$, $t_{thres} = 20$, $K = 5$, $S_2 = 40$ and $q = 25$.

We evaluate the performance of each algorithm by two criteria, loss function value of $\Phi(U)$ and the ratio of distance to the optimum $\|UU^T - M^*\|_F^2/\|M^*\|_F^2$. The experimental results of these two quantities versus the number of gradient oracles are shown in Figure 1.

From the experimental results we can see SGDA, Acc-MDA and SREDA cannot escape saddle points because the loss function value is far away from the global minimum 0, which is equivalent to local minimum in this task because of the strict saddle property. In contrast, we can see our PRGDA algorithm eventually converges to the global optimum $U^*$ and achieves the best loss function value that is close to 0, which indicates its ability to escape saddle point. Especially in the case of $d = 50$, we can see clearly that our PRGDA algorithm jumps out of the trap of saddle point. Besides, in our experiment we also list the smallest eigenvalue of the Hessian matrix $\nabla^2\Phi(U)$ for each algorithm after they have converged. Each algorithm is run for 5 times and the mean value is reported in Table 3. We can see the value $\lambda_{min}(\nabla^2\Phi(U))$ of our method is the closest to 0 in all cases, which also verifies the performance of our PRGDA algorithm to find second-order stationary point.

## 7.2 Hyper-Representation Learning

We conduct a hyper-representation learning experiment to verify the ability of our method to reach second-order stationary point in bilevel optimization. Recently, many methods in meta learning Finn et al. (2017); Nichol & Schulman (2018) are designed to learn hyper-representations via two steps and separated dataset. The backbone is trained to extract better feature representations which can be applied to many different tasks. Based on these features a classifier is further learned on specific type of

Table 3: Smallest eigenvalue of $\nabla^2\Phi(U)$.

| Algorithm | $d = 50$ | $d = 75$ | $d = 100$ |
|---|---|---|---|
| SGDA | -0.0788 | -0.0688 | -0.0360 |
| Acc-MDA | -0.0677 | -0.0420 | -0.0257 |
| SREDA | -0.0746 | -0.0414 | -0.0259 |
| PRGDA | **-0.0018** | **-0.0074** | **-0.0071** |

training data, which eventually forms a bilevel problem. In this experiment we also consider the matrix sensing task but conduct it in the hyper-representation learning manner.

The generation of $U^*$, $M^*$, $A_i$ and $b_i$ are the same as Section 7.1. We also set $d = 50$, $d = 75$ and $d = 100$. The number of samples is $n = 20d$. We split all samples into two dataset: a train dataset $D_1$ with 70% data and a validation dataset $D_2$ with 30% data. We define variable $x$ to be the first $r - 1$ columns of $U$ and variable $y$ to be the last column. The objective function is formulated by

$$\min_{x \in \mathbb{R}^{d \times (r-1)}} \frac{1}{2|D_1|} \sum_{i \in D_1} L_i(x, y^*(x)), \text{ where } y^*(x) = \arg\min_{y \in \mathbb{R}^d} \frac{1}{2|D_2|} \sum_{i \in D_2} L_i(x, y) \quad (11)$$

Here $L_i(\cdot)$ is defined in Eq. (9) since $U$ is the concatenation of $x$ and $y$.

We follow the initialization in Section 7.1 to set $x = [u_0, \mathbf{0}]$ and $y = \mathbf{0}$. We compare our PRGDA algorithm with four baselines, StocBiO, MRBO, VRBO and StocBiO + iNEON. We choose $\eta = 0.001$, $\eta_H = 0.1$, $\lambda = 0.01$, $D = r = 0.01$, $t_{thres} = 20$, $K = 5$, $S_2 = 40$ and $q = 25$. We also use the ration of distance to optimum, *i.e.* $\|UU^T - M^*\|_F^2/\|M^*\|_F^2$ as the metric to evaluate the performance. The experimental results are shown in Figure 2.

The experimental results indicate our PRGDA algorithm shows the best performance to reach second-order stationary point and approach the expected optimum. MRBO and VRBO do not escape saddle points during the experiment. In the case of $d = 50$, StocBiO performs better than MRBO and VRBO because the randomness of stochastic gradient serves as a kind of perturbation, while in variance-reduced algorithms the gradient estimator is closer to the full gradient. This result indicates the necessity of our method to make variance-reduced bilevel algorithm escape saddle points. StocBiO + iNEON also escapes saddle point probably but its convergence is slower than our method.

## 8 Conclusion

In this paper, we propose a new algorithm PRGDA for stochastic nonconvex hierarchical optimization which is the first algorithm to find second-order stationary point for stochastic nonconvex minimax optimization. We prove that our method obtains the gradient complexity of $\tilde{O}(\epsilon^{-3})$ to achieve $O(\epsilon, \sqrt{\rho_\Phi \epsilon})$ second-order stationary point, which matches the best results of searching first-order stationary point under same conditions. We also conduct two numerical experiments, robust optimization and hyper-representation learning to verify the performance of our algorithm.

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

# A DETAILS OF RELATED WORK

## A.1 STOCHASTIC MINIMAX OPTIMIZATION

In recent years, many algorithms for solving stochastic minimax optimization were proposed, and the majority of them were studied under the nonconvex-strongly-concave condition. SGDmax (Jin et al. (2019)) is an intuitive double loop algorithm that extends SGD to minimax problem and achieves SFO complexity of $O(\kappa^3 \epsilon^{-4} log(1/\epsilon))$ where $\kappa$ is the condition number. Stochastic Gradient Descent Ascent (SGDA) (Lin et al. (2020a)) is a single loop algorithm to solve nonconvex-strongly-concave and nonconvex-concave minimax problems. For nonconvex-strongly-concave problem, it requires $O(\kappa^3 \epsilon^{-4})$ SFO complexity to find an $\epsilon$-stationary point of $\Phi(x)$. More recently in (Yang et al. (2022)), a new method Stoc-Smoothed-AGDA is proposed to achieve the SFO complexity of $O(\kappa^2 \epsilon^{-4})$ with a weaker PL condition instead of strong concavity.

Stochastic Recursive gradiEnt Descent Ascent (SREDA) (Luo et al. (2020)) is a double loop algorithm that achieves $O(\kappa^3 \epsilon^{-3})$ SFO complexity which is state-of-the-art of stochastic nonconvex-strongly-concave minimax optimization. It accelerates SGDA by using SPIDER, which is a variance reduction technique and utilizes the newest gradient information (Fang et al. (2018); Nguyen et al. (2017)). SREDA also involves a separated initialization algorithm called PiSARAH (Nguyen et al. (2021)) to ensure the convergence. Hybrid Variance-Reduced SGD (Tran-Dinh et al. (2020)) takes advantage of another variance reduction technique named STORM or hybrid variance reduced stochastic gradient descent (Cutkosky & Orabona (2019)) to accelerate the algorithm for a special case of minimax optimization. Acc-MDA (Huang et al. (2022a)) also uses STORM to realize acceleration for minimax optimization and it achieves the SFO complexity of $O(\kappa^{4.5} \epsilon^{-3})$ without large batches. There are also some works that study the weakly-convex concave minimax optimization such as (Rafique et al. (2021)) and (Yan et al. (2020)). More recently, there are many other works that are proposed to improve the efficiency of stochastic nonconvex minimax optimization algorithms in various aspects, such as adaptive gradient (Guo et al. (2021); Huang & Huang (2021)) and decentralization (Liu et al. (2019); Xian et al. (2021)).

## A.2 CUBIC-GDA AND MINIMAX CUBIC NEWTON

Cubic-Regularized Gradient Descent Ascent (Cubic-GDA) (Chen et al. (2021b)) and Minimax Cubic Newton (MCN) (Luo & Chen (2021)) are two recent algorithms that can reach the second-order stationary point of envelope function $\Phi(x)$ in nonconvex-strongly-concave minimax optimization. Both of these two algorithms are inspired by cubic regularization and designed for deterministic problem. Cubic regularization was first proposed in (Nesterov & Polyak (2006)) which is a standard method that converges to second-order stationary point in conventional nonconvex optimization. Cubic-GDA incorporates cubic regularization with GDA which alternately updates $y$ by gradient descent and updates $x$ following the iterative rule of cubic regularization algorithm. (Chen et al. (2021b)) analyzes the asymptotic convergence rate of Cubic-GDA to guarantee it converges to second-order stationary point eventually.

MCN algorithm is another minimax algorithm based on regularization to find second-order stationary point. It adopts Accelerated Gradient Descent (AGD) (Nesterov (2003)) to update variable $y$ and evaluate the maximum $y^*(x)$. Then it updates $x$ by constructing inexact first-order and second-order information of $\Phi(x)$ and solving the cubic regularized quadratic problem. In (Luo & Chen (2021)) the authors provide the non-asymptotic convergence analysis to show that MCN algorithm requires $\tilde{O}(\kappa^2 \sqrt{\rho}\epsilon^{-1.5})$ first-order oracle calls and $O(\kappa^{1.5}\sqrt{\rho}\epsilon^{-1.5})$ second-order oracle calls or $\tilde{O}(\kappa^{1.5}\epsilon^{-2})$ Hessian vector product calls to achieve $O(\epsilon, \sqrt{\kappa^3 \rho \epsilon})$ second-order stationary point.

As is mentioned, Cubic-GDA and MCN are both designed for deterministic problem and neither of them works for the stochastic minimax problem (2) considered in this paper. Therefore, we are motivated to propose an algorithm that is suitable for the stochastic problem. Besides, Cubic-GDA and MCN involves the calculation of second-order oracle or Hessian vector product while our method only requires the first-order information, which indicates that our method is more efficient to implement because the computation cost of Hessian matrix could be extremely high.

### A.3 PERTURBED GRADIENT DESCENT

Perturbed Gradient Descent (PGD) algorithm (Jin et al. (2017)) was proposed to find second-order stationary point for nonconvex optimization which introduces a perturbation under specific condition. It is a deterministic gradient based algorithm and only involves first-order oracle. It requires $\tilde{O}(\epsilon^{-2})$ gradient oracles to achieve $O(\epsilon, \sqrt{\rho\epsilon})$ second-order stationary point which is the same as vanilla Gradient Descent if hiding logarithm. Perturbed Gradient Descent algorithm consists of two phases, a descent phase and an escaping phase. In the descent phase, the algorithm runs gradient descent to make the function value decrease until the magnitude of the gradient is smaller than a certain threshold. In the escaping phase, it first introduces a perturbation drawn from a uniform distribution on the ball $B_0(r)$ with center $\mathbf{0}$ and radius $r$. After certain iterations of gradient descent, if the function value is reduced by a significant threshold then it indicates that the algorithm escapes a saddle point and it will do the descent phase again. Otherwise, it can be proven that the point where the last descent terminates is second-order stationary with high probability.

Pullback algorithm (Chen et al. (2021a)) extends Perturbed Gradient Descent to the stochastic setting and incorporates it with variance reduction techniques SPIDER (Fang et al. (2018)) and STORM (Cutkosky & Orabona (2019)). It requires $\tilde{O}(\epsilon^{-3} + \epsilon_H^{-6})$ SFO complexity to achieve $O(\epsilon, \epsilon_H)$ second-order stationary points. Different from the deterministic case, perturbed stochastic gradient method in stochastic problem encounters more challenges since the exact objective function value and gradient cannot be accessed. In the escaping phase Pullback determines when to break the phase according to the average moving distance $\bar{D}$. If the accumulated squared moving distance excesses $\bar{D}$, then the approached first-order stationary point is a saddle point with high probability.

## B DESCRIPTION OF ALGORITHM 3

---

**Algorithm 3** Updater of Inner Loop (Bilevel)

---

**Input**: status $x_t$, $x_{t-1}$, $y_t$, $v_{t-1}^{(1)}$, $v_{t-1}^{(2)}$, $u_{t-1}$ and $t$
**Parameter**: stepsize $\lambda$ and $\alpha$, inner loop size $K$ and $Q$, batchsize $B$, $S_1$, $S_2$, $S_3$, $S_4$ and $S_5$, period $q$.

1: Set $x_{t,-1} = x_{t-1}$, $x_{t,k} = x_t$ when $k \geq 0$, $y_{t,-1} = y_{t,0} = y_t$.
2: **if** $mod(t, q) = 0$ **then**
3:   Draw $S_1$ samples $\{\xi_1, \ldots, \xi_{S_1}\}$, $S_3$ samples $\{\zeta_1, \ldots, \zeta_{S_3}\}$
4:   $v_{t,-1}^{(1)} = \frac{1}{S_1} \sum_{i=1}^{S_1} \nabla_x F(x_t, y_t; \xi_i)$, $v_{t,-1}^{(2)} = \frac{1}{S_1} \sum_{i=1}^{S_1} \nabla_y F(x_t, y_t; \xi_i)$,
5:   $u_{t,-1} = \frac{1}{S_3} \sum_{i=1}^{S_3} \nabla_y G(x_t, y_t; \zeta_i)$.
6: **else**
7:   $v_{t,-1}^{(1)} = v_{t-1}^{(1)}$, $v_{t,-1}^{(2)} = v_{t-1}^{(2)}$, $u_{t,-1} = u_{t-1}$.
8: **end if**
9: **for** $k = 0$ **to** $K - 1$ **do**
10:   Draw $S_2$ samples $\{\xi_1, \ldots, \xi_{S_2}\}$, $S_4$ samples $\{\zeta_1, \ldots, \zeta_{S_4}\}$
11:   $v_{t,k}^{(1)} = v_{t,k-1}^{(1)} + \frac{1}{S_2} \sum_{i=1}^{S_2} (\nabla_x F(x_{t,k}, y_{t,k}; \xi_i) - \nabla_x F(x_{t,k-1}, y_{t,k-1}; \xi_i))$
12:   $v_{t,k}^{(2)} = v_{t,k-1}^{(2)} + \frac{1}{S_2} \sum_{i=1}^{S_2} (\nabla_y F(x_{t,k}, y_{t,k}; \xi_i) - \nabla_y F(x_{t,k-1}, y_{t,k-1}; \xi_i))$
13:   $u_{t,k} = u_{t,k-1} + \frac{1}{S_4} \sum_{i=1}^{S_4} (\nabla_y G(x_{t,k}, y_{t,k}; \zeta_i) - \nabla_y G(x_{t,k-1}, y_{t,k-1}; \zeta_i))$
14:   $y_{t,k+1} = \prod_{\mathcal{Y}} (y_{t,k} - \lambda u_{t,k})$.
15: **end for**
16: Select $s_t = \arg\min_k \|\tilde{\mathcal{G}}_\lambda(y_{t,k})\|$. Let $y_{t+1} = y_{t,s_t}$, $v_t^{(1)} = v_{t,s_t}^{(1)}$, $v_t^{(2)} = v_{t,s_t}^{(2)}$, $u_t = u_{t,s_t}$.
17: Compute Jacobian $J_t = \frac{1}{S_5} \sum_{i=1}^{S_5} \nabla_{xy}^2 G(x_t, y_{t+1}; \zeta_i)$.
18: Compute $v_t$ via AID in Eq. (8) based on $v_t^{(1)}$, $v_t^{(2)}$ and $J_t$.
**Output**: $y_{t+1}$, $v_t$, $u_t$.

---

## C ADDITIONAL EXPERIMENTAL RESULTS

In this section we will show some additional results in the robust matrix sensing experiment. We demonstrate the experimental results under different choices of the rank of matrix in Figure 3.

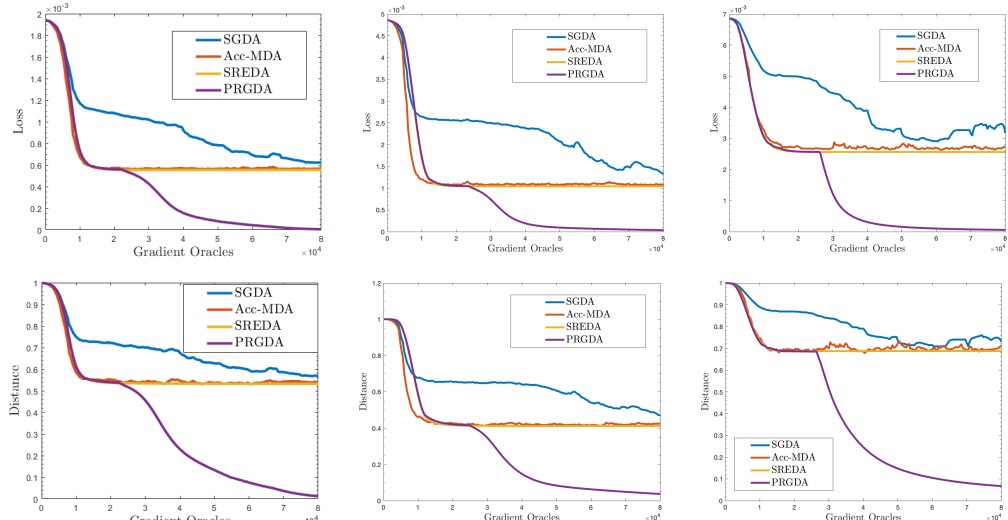

Figure 3: Experimental results of our robust low-rank matrix sensing task. Figure (a) to (c) show the loss function value of $\Phi(U)$ against the number of gradient oracles with $r = 3$, $r = 5$, and $r = 7$ respectively. Figure (d) to (f) show the ratio of distance $\|UU^T - M^*\|_F^2/\|M^*\|_F^2$ against the number of gradient oracles with $r = 3$, $r = 5$, and $r = 7$ respectively.

# D  MINIMAX OPTIMIZATION

## D.1  PROOF SKETCH OF THEOREM 1

First, we define the following notations.

$$\mathcal{G}_\lambda(x, y) = \frac{y - \prod_{\mathcal{Y}}(y + \lambda \nabla_y f(x, y))}{\lambda}, \; \gamma_t = \mathcal{G}_\lambda(x_t, y_{t+1}),$$

$$\epsilon_t = v_t - \nabla_x f(x_t, y_{t+1}), \; \theta_t = u_t - \nabla_y f(x_t, y_{t+1}) \tag{12}$$

Additionally, we assume that each component function $F(x, y; \xi)$ satisfies bounded variance, *i.e.*,

$$\|\nabla F(x, y; \xi) - \nabla f(x, y)\| \leq \sigma \tag{13}$$

Then we have the following estimation of $\epsilon_t$, $\theta_t$ and $\gamma_t$ in Lemma 1 to show their magnitude are bounded by $O(\kappa^{-1}\epsilon)$ and $\|v_t - \nabla\Phi(x_t)\|$ is bounded by $O(\epsilon)$.

**Lemma 1.** *Set stepsize $\eta \leq \frac{\kappa^{-1}\epsilon}{160\log(4/\delta_1)LC_1}$, $\lambda = \frac{1}{6L}$, batchsize $S_2 \geq 819200\log^2(4/\delta_1)\kappa\epsilon^{-1}$, $S_1 \geq 204800\log^2(4/\delta_1)\sigma^2\kappa^2\epsilon^{-2}$, period $q = \epsilon^{-1}$, inner loop $K \geq 2304\kappa$, perturbation radius $r \leq \frac{\kappa^{-1}\epsilon}{160\log(4/\delta_1)LC_1}$ and average movement $\bar{D} \leq \frac{\kappa^{-2}\epsilon^2}{25600\log^2(4/\delta_1)L^2C_1^2}$ where $C_1 = \tilde{O}(1)$ is a constant to be decided later. The initial value of $y_0$ satisfies $\|\mathcal{G}_\lambda(x_0, y_0)\| \leq \frac{\kappa^{-1}\epsilon}{4C_1}$. With probability at least $1 - 4\delta_1$, for $\forall t$ we have $\|\epsilon_t\| \leq \frac{\kappa^{-1}\epsilon}{160C_1}$, $\|\theta_t\| \leq \frac{\kappa^{-1}\epsilon}{160C_1}$ and $\|\gamma_t\| \leq \frac{\kappa^{-1}\epsilon}{4C_1}$. Moreover, we have $\|v_t - \nabla\Phi(x)\| \leq \frac{\epsilon}{C_1}$.*

*Proof.* According to the definition of $\epsilon_t$ and $\theta_t$, when $mod(t + 1, q) \neq 0$ we have

$$\epsilon_{t+1} - \epsilon_t = \frac{1}{S_2} \sum_{k=1}^{s_t} \sum_{i=1}^{S_2} \Big(\nabla_x F(x_{t+1}, y_{t+1,k}; \xi_{k,i}) - \nabla_x F(x_{t+1}, y_{t+1,k-1}; \xi_{k,i})$$

$$- (\nabla_x f(x_{t+1}, y_{t+1,k}) - \nabla_x f(x_{t+1}, y_{t+1,k-1}))\Big) + \frac{1}{S_2} \sum_{i=1}^{S_2} \nabla_x F(x_{t+1}, y_{t+1}; \xi_i)$$

$$- \nabla_x F(x_t, y_{t+1}; \xi_i) - (\nabla_x f(x_{t+1}, y_{t+1}) - \nabla_x f(x_t, y_{t+1})) \tag{14}$$

and

$$\theta_{t+1} - \theta_t = \frac{1}{S_2} \sum_{k=1}^{s_t} \sum_{i=1}^{S_2} \Big( \nabla_y F(x_{t+1}, y_{t+1,k}; \xi_{k,i}) - \nabla_y F(x_{t+1}, y_{t+1,k-1}; \xi_{k,i})$$

$$- (\nabla_y f(x_{t+1}, y_{t+1,k}) - \nabla_y f(x_{t+1}, y_{t+1,k-1})) \Big) + \frac{1}{S_2} \sum_{i=1}^{S_2} \nabla_y F(x_{t+1}, y_{t+1}; \xi_i)$$

$$- \nabla_y F(x_t, y_{t+1}; \xi_i) - (\nabla_y f(x_{t+1}, y_{t+1}) - \nabla_y f(x_t, y_{t+1})) \tag{15}$$

Applying Azuma-Hoeffding inequality (Lemma 7) and union bound, for $\forall t$, with probability at least $1 - 2\delta_1$ we have

$$\|\epsilon_{t+1}\|^2 \le 4 \log(4/\delta_1) \Big( \frac{\sigma^2}{S_1} + \frac{4L^2}{S_2} \sum_{i=\lfloor t/q \rfloor q}^{t} (\|x_{i+1} - x_i\|^2 + \sum_{k=1}^{s_i} \|y_{i+1,k} - y_{i+1,k-1}\|^2) \Big) \tag{16}$$

$$\|\theta_{t+1}\|^2 \le 4 \log(4/\delta_1) \Big( \frac{\sigma^2}{S_1} + \frac{4L^2}{S_2} \sum_{i=\lfloor t/q \rfloor q}^{t} (\|x_{i+1} - x_i\|^2 + \sum_{k=1}^{s_i} \|y_{i+1,k} - y_{i+1,k-1}\|^2) \Big) \tag{17}$$

We define

$$\Delta_{t,k} = \langle y_{t,k} - y_{t,k-1}, u_{t,k} - u_{t,k-1} - (\nabla_y f(x_t, y_{t,k}) - \nabla_y f(x_t, y_{t,k-1})) \rangle$$

$$= \frac{1}{S_2} \sum_{i=1}^{S_2} \langle y_{t,k} - y_{t,k-1}, \nabla_y F(x_t, y_{t,k}; \xi_{k,i}) - \nabla_y F(x_t, y_{t,k-1}; \xi_{k,i})$$

$$- (\nabla_y f(x_t, y_{t,k}) - \nabla_y f(x_t, y_{t,k-1})) \rangle \tag{18}$$

Then by Lemma 8 we can obtain

$$\|y_{t,k+1} - y_{t,k}\|^2$$

$$\le (1 - \frac{2\lambda\mu L}{\mu + L}) \|y_{t,k} - y_{t,k-1}\|^2 - (\frac{2\lambda}{\mu + L} - \lambda^2) \|\nabla_y f(x_t, y_{t,k}) - \nabla_y f(x_t, y_{t,k-1})\|^2$$

$$+ 2\lambda \langle y_{t,k} - y_{t,k-1}, u_{t,k} - u_{t,k-1} - (\nabla_y f(x_t, y_{t,k}) - \nabla_y f(x_t, y_{t,k-1})) \rangle$$

$$\le (1 - \frac{2\lambda\mu L}{\mu + L}) \|y_{t,k} - y_{t,k-1}\|^2 + 2\lambda \Delta_{t,k} \tag{19}$$

Here in the second inequality we use the relation $\lambda \le \frac{1}{L}$. Sum Eq. (19) and we have

$$\sum_{k=1}^{s_t-1} \|y_{t,k+1} - y_{t,k}\|^2 \le (1 - \frac{2\lambda\mu L}{\mu + L}) \sum_{k=0}^{s_t-1} \|y_{t,k+1} - y_{t,k}\|^2 + 2\lambda \sum_{k=1}^{s_t-1} \Delta_{t,k} \tag{20}$$

Moving the first term on the right side of Eq. (20) to the left side and applying Azuma-Hoeffding inequality to $\Delta_{t,k}$, we have

$$\sum_{i=\lfloor t/q \rfloor q}^{t} \sum_{k=1}^{s_i} \|y_{i+1,k} - y_{i+1,k-1}\|^2$$

$$\le \frac{\lambda(\mu + L)}{2\mu L} \sum_{i=\lfloor t/q \rfloor q}^{t} \|\frac{y_{i+1,1} - y_{i+1,0}}{\lambda}\|^2 + \frac{\mu + L}{\mu L} \sum_{i=\lfloor t/q \rfloor q}^{t} \sum_{k=1}^{s_i-1} \Delta_{i+1,k}$$

$$\le \frac{\lambda\kappa}{L} \sum_{i=\lfloor t/q \rfloor q}^{t} \|\frac{y_{i+1,1} - y_{i+1,0}}{\lambda}\|^2 + \frac{(1+\kappa)}{L} \frac{4L \log(4/\delta_1)}{S_2} \sum_{i=\lfloor t/q \rfloor q}^{t} \sum_{k=1}^{s_i-1} \|y_{i+1,k} - y_{i+1,k-1}\|^2$$

$$\tag{21}$$

From Lemma 12 in (Luo et al. (2020)) we know

$$\|\frac{y_{i,1} - y_{i,0}}{\lambda}\|^2 \le 3\|u_{i,0} - \nabla_y f(x_i, y_i)\|^2 + 3L^2 \|x_i - x_{i-1}\|^2 + 3\|\gamma_{i-1}\|^2$$

$$\le 9\|\theta_{i-1}\|^2 + 21L^2 \|x_i - x_{i-1}\|^2 + 3\|\gamma_{i-1}\|^2 \tag{22}$$

In the second inequality we use Cauchy-Schwartz inequality and Assumption 2 since

$$u_{i,0} = u_{i-1} + \frac{1}{S_2} \sum_{j=1}^{S_2} \nabla_y F(x_i, y_i; \xi_{i,j}) - \nabla_y F(x_{i-1}, y_i; \xi_{i,j}) \tag{23}$$

Then by the choice of $S_2 \geq 8\kappa \log(4/\delta_1)$ we can obtain

$$\sum_{i=\lfloor t/q \rfloor q}^{t} \sum_{k=1}^{s_i} \|y_{i+1,k} - y_{i+1,k-1}\|^2 \leq \frac{2\lambda\kappa}{L} \sum_{i=\lfloor t/q \rfloor q}^{t} \|\frac{y_{i+1,1} - y_{i+1,0}}{\lambda}\|^2$$

$$\leq \frac{6\lambda\kappa}{L} \sum_{i=\lfloor t/q \rfloor q}^{t} (3\|\theta_i\|^2 + 7L^2\|x_{i+1} - x_i\|^2 + \|\gamma_i\|^2) \tag{24}$$

Using the choice of $\lambda \leq \frac{1}{6L}$ we can further conclude

$$\|\epsilon_{t+1}\|^2 \leq 4\log(4/\delta_1)\Big(\frac{\sigma^2}{S_1} + \frac{4\kappa}{S_2} \sum_{i=\lfloor t/q \rfloor q}^{t} (8L^2\|x_{i+1} - x_i\|^2 + 3\|\theta_i\|^2 + \|\gamma_i\|^2)\Big) \tag{25}$$

$$\|\theta_{t+1}\|^2 \leq 4\log(4/\delta_1)\Big(\frac{\sigma^2}{S_1} + \frac{4\kappa}{S_2} \sum_{i=\lfloor t/q \rfloor q}^{t} (8L^2\|x_{i+1} - x_i\|^2 + 3\|\theta_i\|^2 + \|\gamma_i\|^2)\Big) \tag{26}$$

Next we will estimate the bound of $\|\gamma_t\|$. Define

$$\bar{y}_{t,k+1} = \Pi_{\mathcal{Y}}(y_{t,k} + \lambda \nabla_y f(x_t, y_{t,k})) \tag{27}$$

Then according to the proof of SREDA (Lemma 10 Eq. (9) in (Luo et al. (2020))), we have

$$f(x_t, y_{t,k}) \leq f(x_t, y_{t,k+1}) - (\frac{1}{2\lambda} - \frac{L}{2})\|y_{t,k+1} - y_{t,k}\|^2 - (\frac{1}{3\lambda} - L)\|\bar{y}_{t,k+1} - y_{t,k}\|^2$$

$$+ \lambda\|u_{t,k} - \nabla_y f(x_t, y_{t,k})\|^2$$

$$\leq f(x_t, y_{t,k+1}) - (\frac{1}{2\lambda} - \frac{L}{2})\|y_{t,k+1} - y_{t,k}\|^2 - (\frac{1}{3\lambda} - L)\|\bar{y}_{t,k+1} - y_{t,k}\|^2$$

$$+ 4\lambda\log(4/\delta_1)(\|u_{t,0} - \nabla_y f(x_t, y_{t,0})\|^2 + \frac{L^2}{S_2} \sum_{i=0}^{k-1} \|y_{t,i+1} - y_{t,i}\|^2) \tag{28}$$

where in the second inequality Azuma-Hoeffding inequality is applied to $\|u_{t,k} - \nabla_y f(x_t, y_{t,k})\|^2$ which is similar to Eq. (17) to get

$$\|u_{t,k} - \nabla_y f(x_t, y_{t,k})\|^2 \leq 4\log(4/\delta_1)(\|u_{t,0} - \nabla_y f(x_t, y_{t,0})\|^2 + \frac{L^2}{S_2} \sum_{i=0}^{k-1} \|y_{t,i+1} - y_{t,i}\|^2) \tag{29}$$

Applying recursion on Eq. (28), for any $k \leq K$ we have

$$f(x_t, y_{t,1}) \leq f(x_t, y_{t,k}) - \sum_{i=1}^{k}(\frac{1}{2\lambda} - \frac{L}{2} - \frac{4L^2\lambda\log(4/\delta_1)}{S_2})\|y_{t,i+1} - y_{t,i}\|^2 - (\frac{1}{3\lambda} - L)$$

$$\cdot \sum_{i=1}^{k} \|\bar{y}_{t,i+1} - y_{t,i}\|^2 + 4k\lambda\log(4/\delta_1)(\|u_{t,0} - \nabla_y f(x_t, y_{t,0})\|^2 + \frac{L^2}{S_2}\|y_{t,1} - y_{t,0}\|^2)$$

$$\leq f(x_t, y_{t,k}) - 2L^2 \sum_{i=1}^{k} \|y_{t,i+1} - y_{t,i}\|^2 - L\lambda^2 \sum_{i=1}^{k} \|\mathcal{G}_\lambda(x_t, y_{t,i})\|^2$$

$$+ 4k\lambda\log(4/\delta_1)(\|u_{t,0} - \nabla_y f(x_t, y_{t,0})\|^2 + \frac{L^2}{S_2}\|y_{t,1} - y_{t,0}\|^2) \tag{30}$$

where we have used $\lambda \leq \frac{1}{6L}$ and the definition of $\mathcal{G}_\lambda(x, y)$. Let $k = K$ we achieve

$$\sum_{k=1}^{K} \|\mathcal{G}_\lambda(x_t, y_{t,k})\|^2 \leq \frac{f(x_t, y^*(x_t)) - f(x_t, y_{t,1})}{L\lambda^2} - \frac{2L}{\lambda^2} \sum_{k=1}^{K} \|y_{t,k+1} - y_{t,k}\|^2$$

$$+ \frac{4K \log(4/\delta_1)}{L\lambda}(\|u_{t,0} - \nabla_y f(x_t, y_{t,0})\|^2 + \frac{L^2}{S_2}\|y_{t,1} - y_{t,0}\|^2) \quad (31)$$

Due to the definition of $\tilde{\mathcal{G}}_\lambda(y_{t,k})$, we have

$$\|\tilde{\mathcal{G}}_\lambda(y_{t,k}) - \mathcal{G}_\lambda(x_t, y_{t,k})\| = \frac{1}{\lambda^2}\|\Pi_{\mathcal{Y}}(y_{t,k} + \lambda u_{t,k}) - \Pi_{\mathcal{Y}}(y_{t,k} + \lambda \nabla_y f(x_t, y_{t,k}))\|^2$$
$$\leq \|u_{t,k} - \nabla_y f(x_t, y_{t,k})\|^2 \quad (32)$$

because of the non-expansion property of projection. Recall the selection of $s_t$. Then by Cauchy-Schwartz inequality, Eq. (29) and $\lambda = \frac{1}{6L}$ we have

$$\|\mathcal{G}_\lambda(x_t, y_{t,s_t})\|^2$$
$$\leq 2\|\tilde{\mathcal{G}}_\lambda(y_{t,s_t})\|^2 + 2\|u_{t,s_t} - \nabla_y f(x_t, y_{t,s_t})\|^2$$
$$\leq \frac{2}{K}\sum_{k=1}^{K}\|\tilde{\mathcal{G}}_\lambda(y_{t,k})\|^2 + 2\|u_{t,s_t} - \nabla_y f(x_t, y_{t,s_t})\|^2$$
$$\leq \frac{4}{K}\sum_{k=1}^{K}(\|\mathcal{G}_\lambda(x_t, y_{t,k})\|^2 + \|u_{t,k} - \nabla_y f(x_t, y_{t,k})\|^2) + 2\|u_{t,s_t} - \nabla_y f(x_t, y_{t,s_t})\|^2$$
$$\leq \frac{4}{K}\sum_{k=1}^{K}\|\mathcal{G}_\lambda(x_t, y_{t,k})\|^2 + 24\log(4/\delta_1)(\|u_{t,0} - \nabla_y f(x_t, y_{t,0})\|^2 + \frac{L^2}{S_2}\|y_{t,1} - y_{t,0}\|^2)$$
$$\leq \frac{144\kappa}{K}\|\mathcal{G}_\lambda(x_t, y_{t,0})\|^2 + (\frac{144\kappa}{K} + 120\log(4/\delta_1))\|u_{t,0} - \nabla_y f(x_t, y_{t,0})\|^2$$
$$+ \frac{120\log(4/\delta_1)L^2}{S_2}\|y_{t,1} - y_{t,0}\|^2 \quad (33)$$

According to Lemma 8 in (Luo et al. (2020)) and Cauchy-Schwartz inequality we have

$$\|\mathcal{G}_\lambda(x_t, y_{t,0})\|^2 \leq 2L^2\|x_t - x_{t-1}\|^2 + 2\|\gamma_{t-1}\|^2 \quad (34)$$

Therefore, combining Eq. (22), Eq. (33) and Eq. (34), for $\forall t$ we can conclude

$$\|\gamma_{t+1}\|^2 \leq (\frac{288\kappa}{K} + \frac{10\log(4/\delta_1)}{S_2})\|\gamma_t\|^2 + (\frac{432\kappa}{K} + 390\log(4/\delta_1))\|\theta_t\|^2$$
$$+ (\frac{1152\kappa}{K} + 750\log(4/\delta_1))L^2\|x_{t+1} - x_t\|^2 \quad (35)$$

Applying union bound, with probability at least $1 - 4\delta_1$, Eq. (25), Eq. (26) and Eq. (35) hold for $\forall t$. In the descent phase we have $\|x_{t+1} - x_t\|^2 \leq \eta^2$. At the perturbation step we have $\|x_{t+1} - x_t\|^2 \leq r^2$. In the escaping phase, on average we have $\|x_{t+1} - x_t\|^2 \leq \bar{D}$. Thus, we have

$$\|x_{t+1} - x_t\|^2 \leq \max\{\eta^2, r^2, \bar{D}\} \leq \frac{\epsilon^2}{25600\log^2(4/\delta_1)\kappa^2 L^2 C_1^2} \quad (36)$$

According to the choices that $q = \epsilon^{-1}$, $K \geq 2304\kappa$, $S_1 \geq 204800\log^2(4/\delta_1)\sigma^2\kappa^2\epsilon^{-2}$ and $S_2 \geq 819200\log^2(4/\delta_1)\kappa\epsilon^{-1}$, by induction we can prove for $\forall t$, the following bounds hold

$$\|\epsilon_t\|^2 \leq \frac{\epsilon^2}{25600\log(4/\delta_1)\kappa^2 C_1^2} \leq \frac{\epsilon^2}{25600\kappa^2 C_1^2} \quad (37)$$

$$\|\theta_t\|^2 \leq \frac{\epsilon^2}{25600\log(4/\delta_1)\kappa^2 C_1^2} \leq \frac{\epsilon^2}{25600\kappa^2 C_1^2} \quad (38)$$

$$\|\gamma_t\|^2 \leq \frac{\epsilon^2}{16\kappa^2 C_1^2} \quad (39)$$

where the case of $t = 0$ is satisfied by the choice of $S_1$ and the PiSARAH initialization $\|\gamma_0\| \leq \|\mathcal{G}_\lambda(x_0, y_0)\| \leq \frac{\epsilon}{4\kappa C_1}$. By Lemma 9 we can further obtain $\|v_t - \nabla\Phi(x)\| \leq \|\epsilon_t\| + 2\kappa\|\gamma_t\| \leq \frac{\epsilon}{C_1}$. □

Next we will show the result of the decreasing of loss function value $\Phi(x)$ in the descent phase.

**Lemma 2.** *In the descent phase, let stepsize $\eta \leq \frac{\epsilon}{2L_\Phi}$. Then for $\forall s$ we have*

$$\Phi(x_{t_s}) - \Phi(x_{m_s}) \geq \frac{(m_s - t_s)\eta\epsilon}{8} \tag{40}$$

*Proof.* Let $\eta_t = \eta/\|v_t\|$, then we have

$$\Phi(x_{t+1}) \leq \Phi(x_t) + \langle \nabla\Phi(x_t), x_{t+1} - x_t \rangle + \frac{L_\Phi}{2}\|x_{t+1} - x_t\|^2$$

$$\leq \Phi(x_t) - (\frac{\eta_t}{2} - \frac{L_\Phi\eta_t^2}{2})\|v_t\|^2 + \frac{\eta_t}{2}\|v_t - \nabla\Phi(x_t)\|^2 \leq \Phi(x_t) - \frac{\eta\epsilon}{8} \tag{41}$$

where the first inequality is derived by Assumption 2, the second inequality is derived by $2\langle a, b \rangle = \|a\|^2 + \|b\|^2 - \|a - b\|^2$, the third inequality is derived by Cauchy-Schwartz inequality and Lemma 9, and the last inequality is derived by Lemma 1 with $C_1 \geq 2$, $\eta \leq \frac{\epsilon}{2L_\Phi}$ and the condition $\|v_t\| \geq \epsilon$. The conclusion of Lemma 2 can be reached by telescoping Eq. (41). $\square$

From Lemma 2 we can see the average descent of $\Phi(x)$ in the descent phase is $O(\eta\epsilon)$. Next we will show when our algorithm converges to a saddle point after the descent phase, *i.e.*, $\lambda_{min}(\nabla^2\Phi(x_{m_s})) \leq -\epsilon_H$, our algorithm will break the escaping phase with high probability.

**Lemma 3.** *Set stepsize $\eta_H \leq \min\{1/8L_\Phi \log(\frac{\eta_H\epsilon_H\sqrt{d}L_\Phi}{C\rho_\Phi\delta_2 r}), 1/4CL_\Phi \log t_{thres}\}$, escaping phase threshold $t_{thres} = 2\log(\frac{\eta_H\epsilon_H\sqrt{d}L_\Phi}{C\rho_\Phi\delta_2 r})/\eta_H\epsilon_H = \tilde{O}(\frac{1}{\eta_H\epsilon_H})$, perturbation radius $r \leq \frac{L_\Phi\eta_H\epsilon_H}{C\rho_\Phi}$ and average movement $\bar{D} \leq L_\Phi^2\eta_H^2\epsilon_H^2/(C^2\rho_\Phi^2 t_{thres}^2)$ where $C = \tilde{O}(1)$. Then for any $s$, if our PRGDA algorithm does not break the escaping phase, then we have $\lambda_{min}(\nabla^2\Phi(x_{m_s})) \geq -\epsilon_H$, with probability at least $1 - 4\delta_1 - \delta_2$.*

*Proof.* Let $\{x_t\}$, $\{x'_t\}$ be two coupled sequences by running PRGDA algorithm from $x_{m_s+1} = x_{m_s} + \xi$ and $x'_{m_s+1} = x_{m_s} + \xi'$ with $x_{m_s+1} - x'_{m_s+1} = r_0\mathbf{e_1}$, where $\xi, \xi' \in B_0(r)$, $r_0 = \frac{\delta_2 r}{\sqrt{d}}$ and $\mathbf{e_1}$ denotes the smallest eigenvector direction of $\nabla^2\Phi(x_{m_s})$. When $\lambda_{min}(\nabla^2\Phi(x_{m_s})) \leq -\epsilon_H$, by Lemma 10 we have

$$\max_{m_s < t \leq m_s + t_{thres}} \{\|x_t - x_{m_s}\|, \|x'_t - x_{m_s}\|\} \geq \frac{L_\Phi\eta_H\epsilon_H}{C\rho_\Phi}$$

with probability as least $1 - 4\delta_1$. Let $\mathcal{S}$ be the set of $x_{m_s+1}$ that will not generate a sequence moving out of the ball with center $x_{m_s}$ and radius $\frac{L\eta_H\epsilon_H}{C\kappa^2\rho}$. Then the projection of $\mathcal{S}$ onto direction $\mathbf{e_1}$ should not be larger than $r_0$. By integration we can calculate volume of ball and stuck region in $d$-dimension and further check that the probability of $x_{m_s+1} \in \mathcal{S}$ is smaller than $\delta_2$ as $\xi$ is drawn from uniform distribution, which is shown in Eq. (42)

$$Pr(x_{m_s+1} \in \mathcal{S}) \leq \frac{r_0 V_{d-1}(r)}{V_d(r)} \leq \frac{\sqrt{d}r_0}{r} \leq \delta_2 \tag{42}$$

where $V_d(r)$ is the volume of $d$-dimension ball with radius $r$. Applying union bound, with probability at least $1 - 4\delta_1 - \delta_2$ we have

$$\exists m_s < t \leq m_s + t_{thres}, \; \|x_t - x_{m_s+1}\| \geq \frac{L_\Phi\eta_H\epsilon_H}{C\rho_\Phi} \tag{43}$$

If the PRGDA algorithm does not break the escaping phase, then for $\forall m_s < t \leq m_s + t_{thres}$ we have

$$\|x_t - x_{m_s+1}\| < \sqrt{(t - m_s)\sum_{i=m_s+1}^{t-1}\|x_{i+1} - x_i\|^2} \leq (t - m_s)\sqrt{\bar{D}} \tag{44}$$

which is derived by Cauchy-Schwartz inequality. By the choice of parameters $t_{thres}$ and $\bar{D}$, we have

$$\|x_t - x_{m_s+1}\| < t_{thres}\sqrt{\bar{D}} \leq \frac{L_\Phi\eta_H\epsilon_H}{C\rho_\Phi} \tag{45}$$

Therefore, when $\lambda_{min}(\nabla^2\Phi(x_{m_s})) \leq -\epsilon_H$, with probability at least $1 - 4\delta_1 - \delta_2$ our PRGDA algorithm will break the escaping phase. $\square$

Finally, we show the following Lemma 4 of localization, which indicates the decreasing value of $\Phi(x)$ in the escaping phase.

**Lemma 4.** *Let $\delta_2$, $t_{thres}$ and $C$ be the same as Lemma 3 and $\delta_1$ be the same as Lemma 1. Set stepsize*

$$\eta_H = \min\{1/320 L_\Phi \log(4/\delta_1) \log(\frac{\eta_H \epsilon_H \sqrt{d} L_\Phi}{C\rho_\Phi \delta_2 r}), 1/4CL_\Phi \log t_{thres}\} = \tilde{O}(\frac{1}{L_\Phi}) \quad (46)$$

*perturbation radius $r = \min\{\frac{L_\Phi \eta_H \epsilon_H}{C\rho_\Phi}, \frac{\epsilon}{640 \log(4/\delta_1) L_\Phi C_1}\}$ and parameter*

$$\bar{D} = \min\{\frac{L_\Phi^2 \eta_H^2 \epsilon_H^2}{C^2 \rho_\Phi^2 t_{thres}^2}, \frac{\epsilon^2}{25600 \log^2(4/\delta_1) L_\Phi^2 C_1^2}\} = \tilde{O}(\frac{\epsilon^2}{L_\Phi^2}) \quad (47)$$

*where $C_1 = 32C \log^2(\frac{\eta_H \epsilon_H \sqrt{d} L_\Phi}{C\rho_\Phi \delta_2 r}) = \tilde{O}(1)$. Notice that in Eq. (47) we have used $\epsilon_H = \sqrt{\rho_\Phi \epsilon}$. Suppose the PRGDA algorithm breaks the escaping phase started at $x_{m_s}$, then we have*

$$\Phi(x_{m_s}) - \Phi(x_{t_{s+1}}) \geq (t_{s+1} - m_s)\frac{\eta_H \epsilon^2}{2C_1^2} \quad (48)$$

*Proof.* Similar to Eq. (41), we have

$$\Phi(x_{t+1}) \leq \Phi(x_t) + \eta_H \|v_t - \nabla\Phi(x_t)\|^2 - (\frac{1}{2\eta_H} - \frac{L_\Phi}{2})\|x_{t+1} - x_t\|^2 \quad (49)$$

since $\eta_t \leq \eta_H$ for all $m_s < t < t_{s+1}$. According to the definition of $r$ and $\bar{D}$, we can see they satisfy the condition in Lemma 1 and Lemma 3. Telescoping the inequality Eq. (49) we obtain

$$\Phi(x_{m_s+1}) - \Phi(x_{t_{s+1}}) \geq \frac{1}{4\eta_H} \sum_{t=m_s+1}^{t_{s+1}-1} \|x_{t+1} - x_t\|^2 - \eta_H \sum_{t=m_s+1}^{t_{s+1}-1} \|v_t - \nabla\Phi(x_t)\|^2$$

$$\geq (t_{s+1} - m_s - 1)(\frac{\bar{D}}{4\eta_H} - \frac{\eta_H \epsilon^2}{C_1^2}) \geq (t_{s+1} - m_s - 1)\frac{\eta_H \epsilon^2}{C_1^2} \quad (50)$$

where the second inequality uses Lemma 1 and the last inequality uses the definition of $\eta_H$, $\bar{D}$ and $C_1$ in that whichever option $\bar{D}$ takes in Eq. (47), the inequality always holds. Since $\|x_{m_s+1} - x_{m_s}\| = r$, from Eq. (41) we have

$$\Phi(x_{m_s+1}) \leq \Phi(x_{m_s}) + (\|v_t - \nabla\Phi(x_t)\| + \frac{L_\Phi}{2}r)r \leq \Phi(x_{m_s}) + \frac{\eta_H \epsilon^2}{2C_1^2} \quad (51)$$

which is obtained by the definition of $r$. Combining Eq. (50), Eq. (51) and the fact that $t_{s+1} - m_s \geq 2$, we have

$$\Phi(x_{m_s}) - \Phi(x_{t_{s+1}}) \geq (t_{s+1} - m_s)\frac{\eta_H \epsilon^2}{2C_1^2} \quad (52)$$

which finishes the proof. $\square$

According to Lemma 2 and 4, the average descent for each step of PRGDA algorithm is

$$\min\{\frac{\eta\epsilon}{8}, \frac{\eta_H \epsilon^2}{2C_1^2}\} = \tilde{O}(\frac{\epsilon^2}{L_\Phi}) = \tilde{O}(\frac{\epsilon^2}{\kappa L}) \quad (53)$$

where we use the choices of $\eta = \tilde{O}(\frac{\epsilon}{L_\Phi}) = \tilde{O}(\frac{\epsilon}{\kappa L})$ and $\eta_H = \tilde{O}(\frac{1}{L_\Phi}) = \tilde{O}(\frac{1}{\kappa L})$. Therefore, the PRGDA algorithm is guaranteed to terminate and the total number of iterations should be bounded by

$$T \leq \tilde{O}(\frac{L_\Phi(\Phi(x_0) - \Phi^*)}{\epsilon^2}) = \tilde{O}(\frac{\kappa L(\Phi(x_0) - \Phi^*)}{\epsilon^2}) \quad (54)$$

The total SFO complexity can be expressed by

$$I + T \cdot S_2 \cdot K + \frac{T}{q} \cdot S_1 \quad (55)$$

where $I$ represents the complexity of the initialization stage which is $\tilde{O}(\kappa^2 \epsilon^{-2})$ according to (Nguyen et al. (2021); Luo et al. (2020)). With the choices of $S_1$, $S_2$, $q$ and $K$ in Theorem 1, we can obtain the total SFO complexity of $\tilde{O}(\kappa^3 \epsilon^{-3})$. Thus, we have finished the proof of Theorem 1.

## D.2 EXTENSION TO FINITE-SUM PROBLEM

The result of Theorem 1 is achieved under the condition that the problem is stochastic. In a special case where the problem has the form of finite sum with $n$ samples, we can also guarantee the convergence of our algorithm. We replace the large mini-batch of size $S_1$ with the full gradient in Algorithm 1. The analysis is similar to Theorem 1 and we only need to keep the relations that $S_2 = \kappa q$ and $q \cdot S_2 \cdot K = S_1 = n$. Hence we omit the proof in this paper.

**Corollary 1.** *For finite-sum problem, when $n \geq \kappa^2$, we set batch size $S_2 = \tilde{O}(\sqrt{n})$, period $q = O(\kappa^{-1}\sqrt{n})$. Other parameters keep the same as the stochastic case. Then our PRGDA algorithm requires $\tilde{O}(n + \kappa^2\sqrt{n}\epsilon^{-2})$ SFO complexity to achieve $O(\epsilon, \sqrt{\kappa^3\rho\epsilon})$ second-order stationary points with high probability. When $n < \kappa^2$, we have $q = 1$, which means $v_t$ and $u_t$ are deterministic and we always have $\|\epsilon_t\| = \|\theta_t\| = 0$. In this case we can set $S_2 = \tilde{O}(1)$ and the total SFO complexity is $\tilde{O}((\kappa^2 + \kappa n)\epsilon^{-2})$ to achieve $O(\epsilon, \sqrt{\kappa^3\rho\epsilon})$ second-order stationary points with high probability.*

# E BILEVEL OPTIMIZATION

In bilevel optimization, the definitions in Eq. (12) should be modified as follows since $\mathcal{Y} = \mathbb{R}^{d_2}$ in the bilevel case .

$$\mathcal{G}_\lambda(x, y) = \nabla_y g(x, y)), \ \gamma_t = \mathcal{G}_\lambda(x_t, y_{t+1}),$$

$$\epsilon_t^{(1)} = v_t^{(1)} - \nabla_x f(x_t, y_{t+1}), \ \epsilon_t^{(2)} = v_t^{(2)} - \nabla_y f(x_t, y_{t+1}), \ \theta_t = u_t - \nabla_y g(x_t, y_{t+1}) \quad (56)$$

Additionally, we assume that each component function $G(x, y; \xi)$ satisfies bounded variance, *i.e.*,

$$\|\nabla G(x, y; \zeta) - \nabla g(x, y)\| \leq \sigma \quad (57)$$

Then we can obtain a similar lemma to Lemma 1 as follows.

**Lemma 5.** *Set stepsize $\eta \leq \frac{\epsilon\min\{1, \frac{L^2}{2\rho M}\}}{320\log(4/\delta_0)\kappa^3 LC_1}$, $\lambda = \frac{1}{6L}$, batchsize $S_2 \geq 819200\log^2(4/\delta_0)\kappa^{-1}\epsilon^{-1}$, $S_1 \geq 819200\log^2(\frac{4}{\delta_0})\sigma^2\kappa^2\epsilon^{-2}\max\{1, \frac{4\rho^2 M^2}{L^4}\}$, $S_3 \geq 819200\log^2(\frac{4}{\delta_0})\sigma^2\kappa^6\epsilon^{-2}\max\{1, \frac{4\rho^2 M^2}{L^4}\}$, $S_4 \geq 819200\log^2(4/\delta_0)\kappa^3\epsilon^{-1}$, period $q = \kappa^2\epsilon^{-1}$, inner loop $K \geq 2304\kappa$, perturbation radius $r \leq \frac{\bar{\epsilon}\min\{1, \frac{L^2}{2\rho M}\}}{320\log(4/\delta_0)\kappa^3 LC_1}$ and average movement $\bar{D} \leq \frac{\epsilon^2\min\{1, \frac{L^4}{4\rho^2 M^2}\}}{102400\log^2(4/\delta_0)\kappa^6 L^2 C_1^2}$ where $C_1 = \tilde{O}(1)$ is a constant to be decided later. The initial value of $y_0$ satisfies $\|\mathcal{G}_\lambda(x_0, y_0)\| \leq \frac{\epsilon\min\{1, \frac{L^2}{2\rho M}\}}{8\kappa^3 C_1}$. With probability at least $1 - 5\delta_0$, for $\forall t$ we have $\|\epsilon_t^{(1)}\| \leq \frac{\kappa^{-1}\epsilon}{320 C_1}$, $\|\epsilon_t^{(2)}\| \leq \frac{\kappa^{-1}\epsilon}{320 C_1}$, $\|\theta_t\| \leq \frac{\kappa^{-3}\epsilon}{320 C_1}$ and $\|\gamma_t\| \leq \frac{\epsilon\min\{1, \frac{L^2}{2\rho M}\}}{8\kappa^3 C_1}$. Moreover, we have $L\|y_{t+1} - y^*(x_t)\| \leq 2\kappa\|\gamma_t\| \leq \frac{\epsilon\min\{1, \frac{L^2}{2\rho M}\}}{4\kappa^2 C_1}$.*

*Proof.* Similar to Lemma 1 we have

$$\epsilon_{t+1}^{(1)} - \epsilon_t^{(1)} = \frac{1}{S_2}\sum_{k=1}^{s_t}\sum_{i=1}^{S_2}\left(\nabla_x F(x_{t+1}, y_{t+1,k}; \xi_{k,i}) - \nabla_x F(x_{t+1}, y_{t+1,k-1}; \xi_{k,i})\right.$$

$$\left. - (\nabla_x f(x_{t+1}, y_{t+1,k}) - \nabla_x f(x_{t+1}, y_{t+1,k-1}))\right) + \frac{1}{S_2}\sum_{i=1}^{S_2}\nabla_x F(x_{t+1}, y_{t+1}; \xi_i)$$

$$- \nabla_x F(x_t, y_{t+1}; \xi_i) - (\nabla_x f(x_{t+1}, y_{t+1}) - \nabla_x f(x_t, y_{t+1})) \quad (58)$$

$$\epsilon_{t+1}^{(2)} - \epsilon_t^{(2)} = \frac{1}{S_2}\sum_{k=1}^{s_t}\sum_{i=1}^{S_2}\left(\nabla_y F(x_{t+1}, y_{t+1,k}; \xi_{k,i}) - \nabla_y F(x_{t+1}, y_{t+1,k-1}; \xi_{k,i})\right.$$

$$\left. - (\nabla_y f(x_{t+1}, y_{t+1,k}) - \nabla_y f(x_{t+1}, y_{t+1,k-1}))\right) + \frac{1}{S_2}\sum_{i=1}^{S_2}\nabla_y F(x_{t+1}, y_{t+1}; \xi_i)$$

$$- \nabla_y F(x_t, y_{t+1}; \xi_i) - (\nabla_y f(x_{t+1}, y_{t+1}) - \nabla_y f(x_t, y_{t+1})) \quad (59)$$

$$\theta_{t+1} - \theta_t = \frac{1}{S_4}\sum_{k=1}^{s_t}\sum_{i=1}^{S_4}\left(\nabla_y G(x_{t+1}, y_{t+1,k}; \zeta_{k,i}) - \nabla_y G(x_{t+1}, y_{t+1,k-1}; \zeta_{k,i})\right.$$

$$- \left( \nabla_y g(x_{t+1}, y_{t+1,k}) - \nabla_y g(x_{t+1}, y_{t+1,k-1}) \right) + \frac{1}{S_4} \sum_{i=1}^{S_4} \nabla_y G(x_{t+1}, y_{t+1}; \zeta_i)$$

$$- \nabla_y G(x_t, y_{t+1}; \zeta_i) - \left( \nabla_y g(x_{t+1}, y_{t+1}) - \nabla_y g(x_t, y_{t+1}) \right) \tag{60}$$

for $mod(t+1, q) \neq 0$. Using Azuma-Hoeffding inequality, with probability at least $1 - 3\delta_0$ we have

$$\|\epsilon_{t+1}^{(1)}\|^2 \leq 4 \log(4/\delta_0) \left( \frac{\sigma^2}{S_1} + \frac{4L^2}{S_2} \sum_{i=\lfloor t/q \rfloor q}^{t} \left( \|x_{i+1} - x_i\|^2 + \sum_{k=1}^{s_i} \|y_{i+1,k} - y_{i+1,k-1}\|^2 \right) \right) \tag{61}$$

$$\|\epsilon_{t+1}^{(2)}\|^2 \leq 4 \log(4/\delta_0) \left( \frac{\sigma^2}{S_1} + \frac{4L^2}{S_2} \sum_{i=\lfloor t/q \rfloor q}^{t} \left( \|x_{i+1} - x_i\|^2 + \sum_{k=1}^{s_i} \|y_{i+1,k} - y_{i+1,k-1}\|^2 \right) \right) \tag{62}$$

$$\|\theta_{t+1}\|^2 \leq 4 \log(4/\delta_0) \left( \frac{\sigma^2}{S_3} + \frac{4L^2}{S_4} \sum_{i=\lfloor t/q \rfloor q}^{t} \left( \|x_{i+1} - x_i\|^2 + \sum_{k=1}^{s_i} \|y_{i+1,k} - y_{i+1,k-1}\|^2 \right) \right) \tag{63}$$

By Eq. (18) to (24), the estimation

$$\sum_{i=\lfloor t/q \rfloor q}^{t} \sum_{k=1}^{s_i} \|y_{i+1,k} - y_{i+1,k-1}\|^2 \leq \frac{6\lambda\kappa}{L} \sum_{i=\lfloor t/q \rfloor q}^{t} \left( 3\|\theta_i\|^2 + 7L^2 \|x_{i+1} - x_i\|^2 + \|\gamma_i\|^2 \right) \tag{64}$$

is still satisfied when $S_4 \geq 8\kappa \log(1/\delta_0)$, where we only need to replace $f$ with $-g$, $u$ with $-u$ and $S_2$ with $S_4$. Using the choice of $\lambda \leq \frac{1}{6L}$ we can further conclude

$$\|\epsilon_{t+1}^{(1)}\|^2 \leq 4 \log(4/\delta_0) \left( \frac{\sigma^2}{S_1} + \frac{4\kappa}{S_2} \sum_{i=\lfloor t/q \rfloor q}^{t} \left( 8L^2 \|x_{i+1} - x_i\|^2 + 3\|\theta_i\|^2 + \|\gamma_i\|^2 \right) \right) \tag{65}$$

$$\|\epsilon_{t+1}^{(2)}\|^2 \leq 4 \log(4/\delta_0) \left( \frac{\sigma^2}{S_1} + \frac{4\kappa}{S_2} \sum_{i=\lfloor t/q \rfloor q}^{t} \left( 8L^2 \|x_{i+1} - x_i\|^2 + 3\|\theta_i\|^2 + \|\gamma_i\|^2 \right) \right) \tag{66}$$

$$\|\theta_{t+1}\|^2 \leq 4 \log(4/\delta_0) \left( \frac{\sigma^2}{S_3} + \frac{4\kappa}{S_4} \sum_{i=\lfloor t/q \rfloor q}^{t} \left( 8L^2 \|x_{i+1} - x_i\|^2 + 3\|\theta_i\|^2 + \|\gamma_i\|^2 \right) \right) \tag{67}$$

We can mimic the steps in Lemma 1 to obtain the estimation of $\gamma_t$ that

$$\|\gamma_{t+1}\|^2 \leq \left( \frac{288\kappa}{K} + \frac{10 \log(4/\delta_0)}{S_4} \right) \|\gamma_t\|^2 + \left( \frac{432\kappa}{K} + 390 \log(4/\delta_0) \right) \|\theta_t\|^2$$

$$+ \left( \frac{1152\kappa}{K} + 750 \log(4/\delta_0) \right) L^2 \|x_{t+1} - x_t\|^2 \tag{68}$$

The difference of $x_{t+1}$ and $x_t$ can be bounded by

$$\|x_{t+1} - x_t\|^2 \leq \max\{\eta^2, r^2, \bar{D}\} \leq \frac{\epsilon^2 \min\{1, \frac{L^4}{4\rho^2 M^2}\}}{102400 \log^2(4/\delta_0) \kappa^6 L^2 C_1^2} \tag{69}$$

According to $S_1 \geq 819200 \log^2(4/\delta_0) \sigma^2 \kappa^2 \epsilon^{-2} \max\{1, \frac{4\rho^2 M^2}{L^4}\}$, $S_2 \geq 819200 \log^2(4/\delta_0) \kappa^{-1} \epsilon^{-1}$, $S_3 \geq 819200 \log^2(4/\delta_0) \sigma^2 \kappa^6 \epsilon^{-2} \max\{1, \frac{4\rho^2 M^2}{L^4}\}$, $S_4 \geq 819200 \log^2(4/\delta_0) \kappa^3 \epsilon^{-1}$, $q = \kappa^2 \epsilon^{-1}$ and $K \geq 2304\kappa$, by induction we can prove for $\forall t$, the following bounds hold

$$\|\epsilon_t^{(1)}\|^2 \leq \frac{\epsilon^2 \min\{1, \frac{L^4}{4\rho^2 M^2}\}}{102400 \log(4/\delta_0) \kappa^2 C_1^2} \leq \frac{\epsilon^2}{102400 \kappa^2 C_1^2} \tag{70}$$

$$\|\epsilon_t^{(2)}\|^2 \leq \frac{\epsilon^2 \min\{1, \frac{L^4}{4\rho^2 M^2}\}}{102400 \log(4/\delta_0) \kappa^2 C_1^2} \leq \frac{\epsilon^2}{102400 \kappa^2 C_1^2} \tag{71}$$

$$\|\theta_t\|^2 \leq \frac{\epsilon^2 \min\{1, \frac{L^4}{4\rho^2 M^2}\}}{102400 \log(4/\delta_0) \kappa^6 C_1^2} \leq \frac{\epsilon^2}{102400 \kappa^6 C_1^2} \tag{72}$$

$$\|\gamma_t\|^2 \leq \frac{\epsilon^2 \min\{1, \frac{L^4}{4\rho^2 M^2}\}}{64 \kappa^6 C_1^2} \tag{73}$$

where the case of $t = 0$ is satisfied by the choice of $S_1$ and the PiSARAH initialization $\|\gamma_0\| \leq \|\mathcal{G}_\lambda(x_0, y_0)\| \leq \frac{\epsilon \min\{1, \frac{L^2}{2\rho M}\}}{8\kappa^3 C_1}$. $\qquad\qquad\qquad\qquad\qquad\qquad\qquad\qquad\qquad\qquad\qquad\qquad\qquad\square$

Next, we can give the estimation of $\|v_t - \nabla\Phi(x_t)\|$.

**Lemma 6.** *Let $\delta_1 = 7\delta_0/4$ where $\delta_0$ is defined in Lemma 5. Let $|\mathcal{B}_j| = BQ(1 - \alpha\mu)^{Q-j}$, $Q = \tilde{O}(\kappa)$ and $B = 512C_1^2 \log(4/\delta_0)M^2\kappa^2\epsilon^{-2}$ Then we have $\|v_t - \nabla\Phi(x_t)\| \leq \frac{\epsilon}{C_1}$ with probability $1 - 4\delta_1$.*

*Proof.* By Eq. (4) we have

$$v_t - \nabla\Phi(x_t)$$
$$= (v_t^{(1)} - J_t z_t^Q) - (\nabla_x f(x_t, y^*(x_t)) - \nabla_{xy}^2 g(x_t, y^*(x_t))[\nabla_y^2 g(x_t, y^*(x_t))]^{-1}\nabla_y f(x_t, y^*(x_t)))$$
$$= v_t^{(1)} - \nabla_x f(x_t, y_{t+1}) + \nabla_x f(x_t, y_{t+1}) - \nabla_x f(x_t, y^*(x_t)) - (J_t - \nabla_{xy}^2 g(x_t, y^*(x_t)))$$
$$\cdot [\nabla_y^2 g(x_t, y^*(x_t))]^{-1}\nabla_y f(x_t, y^*(x_t)) - J_t(z_t^Q - [\nabla_y^2 g(x_t, y^*(x_t))]^{-1}\nabla_y f(x_t, y^*(x_t))) \quad (74)$$

As $S_5 \geq 64C_1^2 \log^2(4/\delta_0)M^2\kappa^2\epsilon^{-2}$, by Azuma-Hoeffding inequality we have

$$\|J_t - \nabla_{xy}^2 g(x_t, y^*(x_t))\| \leq \frac{L\epsilon}{8\kappa MC_1} \quad (75)$$

with probability $1 - \delta_0$. According to Lemma 5 we have

$$\|v_t - \nabla\Phi(x_t)\| \leq \|\epsilon_t^{(1)}\| + L\|y_{t+1} - y^*(x_t)\| + \frac{M}{\mu}\|J_t - \nabla_{xy}^2 g(x_t, y^*(x_t))\|$$
$$+ L\|z_t^Q - [\nabla_y^2 g(x_t, y^*(x_t))]^{-1}\nabla_y f(x_t, y^*(x_t))\|$$
$$\leq \frac{\epsilon}{2C_1} + L\|z_t^Q - [\nabla_y^2 g(x_t, y^*(x_t))]^{-1}\nabla_y f(x_t, y^*(x_t))\| \quad (76)$$

Next, we will estimate $\|z_t^Q - [\nabla_y^2 g(x_t, y^*(x_t))]^{-1}\nabla_y f(x_t, y^*(x_t))\|$. First, we have

$$\|[\nabla_y^2 g(x_t, y_{t+1})]^{-1}\nabla_y f(x_t, y_{t+1}) - [\nabla_y^2 g(x_t, y^*(x_t))]^{-1}\nabla_y f(x_t, y^*(x_t))\|$$
$$= \|([\nabla_y^2 g(x_t, y_{t+1})]^{-1} - [\nabla_y^2 g(x_t, y^*(x_t))]^{-1})\nabla_y f(x_t, y_{t+1})$$
$$+ [\nabla_y^2 g(x_t, y^*(x_t))]^{-1}(\nabla_y f(x_t, y_{t+1}) - \nabla_y f(x_t, y^*(x_t)))\|$$
$$\leq \frac{\rho M}{\mu^2}\|y_{t+1} - y^*(x_t)\| + \frac{L}{\mu}\|y_{t+1} - y^*(x_t)\| = (\frac{\rho M}{\mu^2} + \kappa)\|y_{t+1} - y^*(x_t)\| \quad (77)$$

We also have estimation

$$\|z_t^Q - [\nabla_y^2 g(x_t, y_{t+1})]^{-1}\nabla_y f(x_t, y_{t+1})\|$$
$$= \|(\alpha \sum_{q=-1}^{Q-1} \prod_{j=Q-q}^{Q} (I - \alpha\nabla_y^2 G(x_t, y_{t+1}; \mathcal{B}_j)) - [\nabla_y^2 g(x_t, y_{t+1})]^{-1})v_t^{(2)}$$
$$+ [\nabla_y^2 g(x_t, y_{t+1})]^{-1}(v_t^{(2)} - \nabla_y f(x_t, y_{t+1}))\|$$
$$\leq 2M\|\alpha \sum_{q=-1}^{Q-1} \prod_{j=Q-q}^{Q} (I - \alpha\nabla_y^2 G(x_t, y_{t+1}; \mathcal{B}_j)) - [\nabla_y^2 g(x_t, y_{t+1})]^{-1}\| + \frac{1}{\mu}\|\epsilon_t^{(2)}\| \quad (78)$$

The first term can be estimated by

$$\|\alpha \sum_{q=0}^{Q} (I - \alpha\nabla_y^2 g(x_t, y_{t+1}))^q - [\nabla_y^2 g(x_t, y_{t+1})]^{-1}\| \leq \alpha \sum_{q=Q+1}^{+\infty} (1 - \alpha\mu)^q = \frac{(1 - \alpha\mu)^{Q+1}}{\mu}$$
$$(79)$$

When $|\mathcal{B}_j| = BQ(1 - \alpha\mu)^{Q-j}$, by Azuma-Hoeffding inequality and the proof of proposition 3 in (Ji et al. (2021)), we have

$$\|\alpha \sum_{q=-1}^{Q-1} \prod_{j=Q-q}^{Q} (I - \alpha\nabla_y^2 G(x_t, y_{t+1}; \mathcal{B}_j)) - \alpha \sum_{q=0}^{Q} (I - \alpha\nabla_y^2 g(x_t, y_{t+1}))^q\|^2$$

$$\leq \frac{\alpha^2 \kappa^2 \log(4/\delta_0)}{B(1-\alpha\mu)} \leq \frac{2\alpha^2 \kappa^2 \log(4/\delta_0)}{B} \tag{80}$$

with probability $1 - \delta_0$. In the second inequality we use $\alpha = \frac{1}{2L}$. Combine Eq. (77) to (80) and we can obtain

$$L\|z_t^Q - [\nabla_y^2 g(x_t, y^*(x_t))]^{-1}\nabla_y f(x_t, y^*(x_t))\|$$

$$\leq 2\kappa^2(1 + \frac{\rho M}{L^2}\kappa)\|\gamma_t\| + \kappa\|\epsilon_t^{(2)}\| + 2\kappa M(1-\alpha\mu)^{Q+1} + \kappa M\sqrt{\frac{2\log(4/\delta_0)}{B}} \leq \frac{\epsilon}{2C_1} \tag{81}$$

where we have used Lemma 5 and the choices of $Q = \tilde{O}(\kappa)$ and $B = 512C_1^2 \log(4/\delta_0)M^2\kappa^2\epsilon^{-2}$ in the last inequality. Therefore, by union bound we have

$$\|v_t - \nabla\Phi(x_t)\| \leq \frac{\epsilon}{C_1} \tag{82}$$

with probability $1 - 7\delta_0$. $\qquad\square$

Now we have reached the same conclusion as the case of minimax optimization. The rest part of the proof for Theorem 2 is almost the same as Theorem 1 since in Lemma 2 to Lemma 4 and Lemma 10 we do not need the specific expression of $v_t$, $L_\Phi$ or $\rho_\Phi$. We only use the bound for $\|v_t - \nabla\Phi(x_t)\|$. The only thing different is that we have to check if $r$ and $\bar{D}$ in Lemma 4 satisfy the condition in Lemma 5, which is affirmative as $L_\Phi \geq \frac{\kappa^3 \rho M}{L}$. Therefore, the average descent is $\tilde{O}(\frac{\epsilon^2}{L_\Phi})$ and $T = \tilde{O}(\frac{L_\Phi}{\epsilon^2}) = \tilde{O}(\kappa^3\epsilon^{-2})$. Finally, we have

$$Gc(f, \epsilon) = TS_2K + \frac{T}{q}S_1 = \tilde{O}(\kappa^3\epsilon^{-3}),\ Gc(g, \epsilon) = I + TS_4K + \frac{T}{q}S_3 = \tilde{O}(\kappa^7\epsilon^{-2}) \tag{83}$$

$$JV(G, \epsilon) = TS_5 = \tilde{O}(\kappa^5\epsilon^{-4}),\ HV(G, \epsilon) = T\sum_{j=0}^{Q-1} BQ(1-\alpha\mu)^j = \frac{TBQ}{\alpha\mu} = \tilde{O}(\kappa^6\epsilon^{-4}) \tag{84}$$

# F AUXILIARY PROPOSITIONS AND LEMMAS

In this section we provide some auxiliary propositions and lemmas used in the proof.

**Proposition 1.** *(Lemma 4.3 in (Lin et al. (2020a))) Suppose function $f$ satisfies Assumption 2 and Assumption 1. Then function $y^*(x)$ is $\kappa$-Lipschitz continuous, i.e.,*

$$\|y^*(x_1) - y^*(x_2)\| \leq \kappa\|x_1 - x_2\|$$

*for $\forall x_1, x_2 \in \mathbb{R}^{d_1}$. Function $\Phi(x)$ is differentiable with gradient $\nabla\Phi(x) = \nabla_x f(x, y^*(x))$ and the gradient is $L_\Phi$-Lipschitz continuous where $L_\Phi = L + \kappa L$.*

**Proposition 2.** *(Lemma 2, Lemma 3 in (Luo & Chen (2021))) Suppose function $f$ satisfies Assumption 2 to Assumption 1. Then function $\Phi(x)$ is twice differentiable and the Hessian is $\rho_\Phi$-Lipschitz continuous where $\rho_\Phi = 4\sqrt{2}\kappa^3\rho$.*

**Proposition 3.** *(Lemma 2.2 in (Ghadimi & Wang (2018))) Under Assumptions 1 to 3, the gradient of $\Phi(x)$ is $L_\Phi$-Lipschitz continuous and the Lipschitz constant $L_\Phi = O(\kappa^3)$ with formula*

$$L_\Phi = L + \frac{2L^2 + \rho M}{\mu} + \frac{L^3 + 2L\rho M}{\mu^2} + \frac{L^2\rho M}{\mu^3}. \tag{85}$$

**Proposition 4.** *(Lemma 3.4 in (Huang et al. (2022b))) Under Assumptions 1 to 5, the Hessian of $\Phi(x)$ is $\rho_\Phi$-Lipschitz continuous and the Lipschitz constant $\rho_\Phi = O(\kappa^5)$.*

Next, we will present the Azuma-Hoeffding inequality.

**Lemma 7.** *(Lemma D.1 in (Chen et al. (2021a))) Let $\epsilon_{1:k} \in \mathbb{R}^d$ be a vector-valued martingale difference sequence with respect to $\mathcal{F}_k$, i.e., for each $k \in [K]$, $\mathbb{E}[\epsilon_k|\mathcal{F}_k] = 0$ and $\|\epsilon_k\| \leq B_k$, then with probability $1 - \delta$ we have*

$$\|\sum_{k=1}^{K} \epsilon_k\|^2 \leq 4\log(4/\delta)\sum_{k=1}^{K} B_k^2 \tag{86}$$

Next we will introduce some lemmas from the convergence analysis of SREDA.

**Lemma 8.** *(Lemma 2 in (Luo et al. (2020))) Suppose $f$ is a $\mu$-strongly convex function and has $L$-Lipschitz gradient. Then for any $x$ and $x'$ we have*

$$\langle \nabla f(x) - \nabla f(x'), x - x' \rangle \geq \frac{\mu L}{\mu + L} \|x - x'\|^2 + \frac{1}{\mu + L} \|\nabla f(x) - \nabla f(x')\|^2 \tag{87}$$

**Lemma 9.** *(Corollary 1 in (Luo et al. (2020))) For any $y \in \mathcal{Y}$ we have*

$$\frac{\mu}{2} \|y - y^*(x_t)\| \leq \|\mathcal{G}_\lambda(x_t, y)\| \tag{88}$$

As $\Phi(x)$ has $(L + \kappa L)$-Lipschitz gradient and $(4\sqrt{2}\kappa^3\rho)$-Lipschitz Hessian, similar to Lemma D.3 in (Chen et al. (2021a)) and Lemma 6 in (Li (2019)) we have the following Lemma 10.

**Lemma 10.** *Set stepsize $\eta_H \leq \min\{1/8L_\Phi \log(\frac{\eta_H \epsilon_H L_\Phi}{C\rho_\Phi r_0}), 1/4CL_\Phi \log t_{thres}\} = \tilde{O}(\frac{1}{\kappa L})$, perturbation radius $r \leq \frac{L_\Phi \eta_H \epsilon_H}{C\rho_\Phi}$ and threshold $t_{thres} = 2\log(\frac{\eta_H \epsilon_H L_\Phi}{C\rho_\Phi r_0})/\eta_H \epsilon_H = \tilde{O}(\frac{1}{\eta_H \epsilon_H})$, where $r_0 \leq r$ and $C = \tilde{O}(1)$. Suppose $-\gamma = \lambda_{min}(\nabla^2\Phi(x_{m_s})) \leq -\epsilon_H$. Let $\{x_t\}$, $\{x'_t\}$ be two coupled sequences by running PRGDA from $x_{m_s+1} = x_{m_s} + \xi$ and $x'_{m_s+1} = x_{m_s} + \xi'$ with $x_{m_s+1} - x'_{m_s+1} = r_0\mathbf{e_1}$, where $\xi, \xi' \in B_0(r)$ and $\mathbf{e_1}$ denotes the smallest eigenvector direction of $\nabla^2\Phi(x_{m_s})$. Then with probability at least $1 - 4\delta_1$ (for $\delta_1$ in Lemma 1), we have*

$$\max_{m_s < t \leq m_s + t_{thres}} \{\|x_t - x_{m_s}\|, \|x'_t - x_{m_s}\|\} \geq \frac{L_\Phi \eta_H \epsilon_H}{C\rho_\Phi} \tag{89}$$

*Proof.* To prove this lemma, we assume the contrary.

$$\forall m_s < t \leq m_s + t_{thres}, \ \|x_t - x_{m_s}\| < \frac{L_\Phi \eta_H \epsilon_H}{C\rho_\Phi}, \ \|x'_t - x_{m_s}\| < \frac{L_\Phi \eta_H \epsilon_H}{C\rho_\Phi} \tag{90}$$

Define $w_t = x_t - x'_t$ and $\nu_t = v_t - \nabla\Phi(x_t) - (v'_t - \nabla\Phi(x'_t))$. We have

$$w_{t+1} = w_t - \eta_H(v_t - v'_t) = w_t - \eta_H(\nabla\Phi(x_t) - \nabla\Phi(x'_t)) - \eta_H\nu_t$$
$$= (I - \eta_H\mathcal{H})w_t - \eta_H(\Delta_t w_t + \nu_t) \tag{91}$$

where

$$\mathcal{H} = \nabla^2\Phi(x_{m_s}), \ \Delta_t = \int_0^1 [\nabla^2\Phi(x'_t + \theta(x_t - x'_t)) - \mathcal{H}]d\theta \tag{92}$$

Let

$$p_{t+1} = (I - \eta_H\mathcal{H})^{t-m_s} w_{m_s+1}, \ q_{t+1} = \eta_H \sum_{\tau=m_s+1}^{t} (I - \eta_H\mathcal{H})^{t-\tau}(\Delta_\tau w_\tau + \nu_\tau) \tag{93}$$

and apply recursion to Eq. (91), we can obtain

$$w_{t+1} = p_{t+1} - q_{t+1} \tag{94}$$

Next, we will inductively prove

$$\|q_t\| \leq \|p_t\|/2, \ \forall m_s < t \leq m_s + t_{thres} \tag{95}$$

First, when $t = m_s + 1$ the conclusion holds since $\|q_{m_s+1}\| = 0$. Suppose Eq. (95) is satisfied for $\tau \leq t$. Then we have

$$\|w_\tau\| \leq \|p_\tau\| + \|q_\tau\| \leq \frac{3}{2}\|p_\tau\| = \frac{3}{2}(1 + \eta_H\gamma)^{\tau-m_s-1}r_0 \tag{96}$$

Then for the case $\tau = t + 1$, by Eq. (93) and (96) we have

$$\|q_{t+1}\| \leq \eta_H(1 + \eta_H\gamma)^{t-m_s}\frac{3}{2}\sum_{\tau=m_s+1}^{t} \|\Delta_\tau\|r_0 + \eta_H\sum_{\tau=m_s+1}^{t}(1 + \eta_H\gamma)^{t-\tau}\|\nu_\tau\|$$

$$\leq (1 + \eta_H\gamma)^{t-m_s}(\frac{3L_\Phi \eta_H^2 \epsilon_H t_{thres}}{2C}r_0 + \frac{1}{4}r_0)$$

$$\leq \frac{1}{2}(1 + \eta_H \gamma)^{t-m_s} r_0 = \|p_{t+1}\|/2 \qquad (97)$$

which finishes the induction of Eq. (95). In the second inequality, we use Lipschitz Hessian and Eq. (90) to obtain $\|\Delta_\tau\| \leq L_\Phi \eta_H \epsilon_H / C$ and we use Lemma 1 in minimax problem or Lemma 5 in bilevel problem and the fact $a^{t+1} - 1 = (a-1) \sum_{s=0}^t a^s$ to obtain $\|\nu_\tau\| \leq \epsilon_H r_0 / 4$ with probability $1 - 4\delta_1$ by choosing constant $C_1 \geq \frac{8\epsilon}{\epsilon_H r_0}$. The last inequality can be achieved by the definitions of $\eta_H$ and $t_{thres}$. Now we have

$$\frac{1}{2}(1 + \eta_H \gamma)^{t-m_s-1} r_0 \leq \|w_t\| \leq \|x_t - x_{m_s}\| + \|x'_t - x_{m_s}\| \qquad (98)$$

which conflicts with Eq. (90) due to the choice of $t_{thres}$. $\qquad\square$

