# OpenReview forum: "Escaping Saddle Point Efficiently in Minimax and Bilevel Optimizations"
_ICLR.cc/2025/Conference — ICLR 2025 Conference Withdrawn Submission_

### Official Review · Reviewer_JSog · 2024-10-28

**Soundness:** 2
**Presentation:** 2
**Contribution:** 3
**Rating:** 3
**Confidence:** 3

**Summary:**

This paper investigates a method called PRGDA, which achieves second-order optimality for nonconvex minimax and bilevel optimization without computing the second-order derivative of the primal function. In nonconvex-strongly-concave minimax optimization, the authors prove that the proposed algorithm can find a second-order stationary point with a gradient complexity that matches state-of-the-art results for finding the first-order stationary point.

**Strengths:**

This paper explores a method that achieves second-order optimality for nonconvex minimax and bilevel optimization without requiring the computation of the second-order derivative of the primal function. It introduces the first stochastic algorithm that is guaranteed to reach the second-order stationary point for nonconvex minimax problems.

**Weaknesses:**

1. My primary concern is the presentation of this paper. For instance:

1-1. In line 230, the author should provide references to support the assertion: "According to previous work $\cdots \cdots$".

1-2. In line 250, the meaning of $v_t$ is unclear. It should be clarified that $v_t$ represents the estimation of the hypergradient and not the partial derivative of $f$ with respect to $x$.

1-3. The purpose of step 13 in Algorithm 1 is unclear and requires clarification.

1-4. In Theorems 1 and 2, the choices for step sizes $\eta$, $\eta_H$, batch sizes $S_1$, $S_2$, period $\rho$, inner loop $K$, perturbation radius $r$, threshold $t_{thres}$, and average movement $\bar{D}$ should be clearly outlined.

1-5. In line 360, definitions for $JV$ and $HV$ should be provided before using them.

2. My primary concern is that the main theorems of this paper (Theorems 1 and 2), which are based on random settings, lack explicit assumptions regarding randomness. Although the authors introduce an assumption about the randomness of the gradient (Equation (13)), I believe such assumptions may be insufficient. It might be necessary to include assumptions about the randomness of function values as well. Furthermore, these assumptions should be included in the main text rather than relegated to the appendix.

3. A comprehensive analysis should compare the complexity results and the adopted assumptions with those of existing methods (Theorems 1 and 2 of this paper).

**Questions:**

1. The definition of hypergradient in Equation (4) should reference earlier works, such as [1].

2. It is essential to reference studies that utilize Assumptions 4 and 5.

3. Equation (8) may require significant computational resources to calculate the hypergradient estimator. Is it possible to replace it with alternative strategies?

4. What does "SFO" signify in lines 180 and 251? It is presumed to denote "stochastic first order."

5. The authors state that "The code of our algorithms is uploaded in the Supplementary Material." in line 375. However, I was unable to locate the Supplementary Material in the submitted documents, which may be due to an issue with my webpage display.

6. For other questions, please see the weaknesses above.


[1] Ghadimi, Saeed, and Mengdi Wang. "Approximation methods for bilevel programming." arxiv preprint arxiv:1802.02246 (2018).

---

### Official Review · Reviewer_uruK · 2024-11-02

**Soundness:** 3
**Presentation:** 2
**Contribution:** 2
**Rating:** 5
**Confidence:** 3

**Summary:**

This paper introduces the Perturbed Recursive Gradient Descent Ascent (PRGDA) algorithm, a novel stochastic method designed to find second-order stationary points in nonconvex minimax optimization problems without requiring second-order derivatives. The algorithm achieves complexity that matches the best-known results for finding first-order stationary points. Additionally, it is applicable to nonconvex bilevel optimization problems.

**Strengths:**

1. PRGDA is the first stochastic algorithm that is guaranteed to obtain the second-order stationary point for nonconvex-strongly-concave minimax problems.
2. PRGDA can be applied to nonconvex bilevel optimization, achieving improved gradient complexity.
3. The authors conduct numerical experiments, including a matrix sensing task, to validate the performance of PRGDA, showcasing its ability to escape saddle points and effectively converge to optimal solutions.

**Weaknesses:**

1. The overall presentation of the article is not clear enough.
(1) In Sections 1 and 2, the author mentions numerous algorithms but does not provide a clear classification of them. For instance, the author could consider classifying the algorithms into deterministic and stochastic categories, and further subdividing them into those capable of finding first-order stationary points and those that can identify second-order stationary points.
(2) In Sections 4 and 5, the author relies solely on textual descriptions and inline formulas, resulting in a dense presentation that can be challenging to understand. It is advisable for the author to condense the text and appropriately position formulas between lines of the text. If there are concerns about exceeding the article's page limit, I recommend focusing on either the minimax problem or the bilevel problem in the main text, with the other problem placed in an appendix.
(3) Compared to existing algorithms, it is unclear where the author's novelty lies.
(4) The author does provide commentary on some existing work, but these observations are scattered throughout the article. It is recommended that the author include more structured 'Remarks'  to enhance readability.

**Questions:**

1. Lines 20-22 and elsewhere: The statement should be, ``... obtain an approximate second-order stationary point with high probability." The paper overstates its results.
2. Line 151: The authors assert that their algorithm is not a nested algorithm. Could they please provide further clarification on this point?
3. Lines 179-180:  The reference to SSRGD is not standardized, and the term 'SFO' is undefined
4. Line 226, Assumption 3：  Does this assumption pertain to one of the variables $x$ or $y$, or does it apply to the pair $(x,y)$?
5. Line 235, Assumption 4: Is this assumption reasonable? Please provide some examples to illustrate.
6. Line 262: Is $\xi$ the same stochastic variable as in equation (2)? Please clarify.
7. Lines 266-267: Why does the algorithm require a pullback? Additionally, "line 17" in the text should be replaced as "line 14".
8. Line 282: The notion of $~B_0(r)$ is misleading, as $B_0(r)$ represents a ball, not a distribution.
9. Theorem 2: "JV" and "HV" are undefined. For bilevel optimization, is the authors' algorithm not classified as a first-order method?
10. Theorems 1 and 2: Why is the maximum number of iterations $T$ for Algorithm 1 not mentioned in Theorems 1 and 2?
11. In the experimental section, the testing problems have not been verified to meet Assumptions 1-3.

---

### Official Review · Reviewer_Z27y · 2024-11-03

**Soundness:** 2
**Presentation:** 2
**Contribution:** 2
**Rating:** 5
**Confidence:** 3

**Summary:**

This paper studies first-order methods for solving non-convex strongly-concave (convex) minimax (bilevel) problems.
The authors establish the complexity of $\tilde{O}(\kappa^3\epsilon^{-3})$ and $\tilde{O}(\kappa^7\epsilon^{-3})$ for finding second-order stationary point in minimax and bilevel optimization respectively.

**Strengths:**

The proposed methods establish new state-of-the-art complexity of using stochastic first-order methods to find the second-order stationary point for NC-SC minimax and bilevel optimization.

**Weaknesses:**

1. Table 1 missed some results, Chen et al. also provides stochastic version of their algorithms.

2. The technique seems quite standard and direct, using existing state-of-the-art variance reduced methods for NC-SC problems and the perturbed methods to escape the saddle point of $\phi(\cdot)$. It is quite unclear what is the technical difficulty used in this paper.

3. This paper studies stochastic problems, however, I do not see any assumptions on the variance of the noise in the main theorems or the assumption section.

**Questions:**

Please refer to the weakness part.

---

### Official Review · Reviewer_6Cu4 · 2024-11-04

**Soundness:** 2
**Presentation:** 2
**Contribution:** 3
**Rating:** 5
**Confidence:** 4

**Summary:**

This paper proposed PRGDA, which is the first algorithm that can find the second-order stationary point for stochastic nonconvex minimax problems, and achieves better gradient complexity to find the second-order stationary point for stochastic nonconvex-strongly-convex bilevel optimization problems. Experiments results in this paper also verified the effectiveness of their proposed algorithm.

**Strengths:**

(1): This paper proposed the first algorithm that can find the second-order stationary point for stochastic nonconvex minimax problems, and proved the complexity matches the best result of finding the first-order stationary point in the same setting.

(2): For finding the second-order stationary point for stochastic nonconvex-strongly-convex bilevel optimization, this paper's algorithm improves the gradient complexity over existing work.

**Weaknesses:**

(1): Usually, for stochastic optimization, we need assumptions regarding noise. However, in the main text and in Theorems 1 and 2, the authors did not state those assumptions. The authors state that their Theorems 1 and 2 actually need additional assumptions in appendix (13) and (57): $||\nabla F(x, y,\xi)-F(x,y)||\leq \delta$ and called it as "bounded variance". However, this is not "bounded variance" and is stronger than the bounded variance assumption that is usually used in stochastic optimization. Could the authors explain why they use this stronger assumption?

(2): The authors never define HV and JV in Theorem 2. Could the authors add the definitions?
I assume they are the required numbers of Hessian and Jacobian. Then, in Theorem 2, the authors showed that their algorithm actually needs $O(\epsilon^{-4})$ Hessian and Jacobian. Thus, the actual complexity of their algorithm for bilevel optimization should be $O(\epsilon^{-4})$. If so, the claimed improvement in complexity over previous work and the claimation that their work matches $O(\epsilon^{-3})$ the best result of finding the first-order stationary point would not hold.

(3): The authors said in the abstract: "Specifically, we propose a new algorithm named PRGDA without the computation of second order derivative of the primal function." However, PRGDA for bilevel optimization actually uses the second order derivatives.

**Questions:**

See Weakness.

---

### Note · Authors · 2025-01-16

I have read and agree with the venue's withdrawal policy on behalf of myself and my co-authors.